# Two peptides targeting endothelial receptors are internalized into murine brain endothelial cells

Diána Hudecz[1][☯], Sara Björk Sigurdardóttir[2][☯], Sarah Christine Christensen[1], Casper Hempel[2], Andrew J. Urquhart[2], Thomas Lars Andresen[2], Morten S. Nielsen[1]*

1 Department of Biomedicine, Faculty of Health, Aarhus University, Aarhus, Denmark, 2 Department of Health Technology, Technical University of Denmark, Lyngby, Denmark

☯ These authors contributed equally to this work.
* mn@biomed.au.dk

**Data Availability Statement:** All relevant data are within the manuscript and its Supporting Information files.

## Abstract

The blood-brain barrier (BBB) is one of the main obstacles for therapies targeting brain diseases. Most macromolecules fail to pass the tight BBB, formed by brain endothelial cells interlinked by tight junctions. A wide range of small, lipid-soluble molecules can enter the brain parenchyma via diffusion, whereas macromolecules have to transcytose via vesicular transport. Vesicular transport can thus be utilized as a strategy to deliver brain therapies. By conjugating BBB targeting antibodies and peptides to therapeutic molecules or nanoparticles, it is possible to increase uptake into the brain. Previously, the synthetic peptide GYR and a peptide derived from melanotransferrin (MTfp) have been suggested as candidates for mediating transcytosis in brain endothelial cells (BECs). Here we study uptake, intracellular trafficking, and translocation of these two peptides in BECs. The peptides were synthesized, and binding studies to purified endocytic receptors were performed using surface plasmon resonance. Furthermore, the peptides were conjugated to a fluorophore allowing for live-cell imaging studies of their uptake into murine brain endothelial cells. Both peptides bound to low-density lipoprotein receptor-related protein 1 (LRP-1) and the human transferrin receptor, while lower affinity was observed against the murine transferrin receptor. The MTfp showed a higher binding affinity to all receptors when compared to the GYR peptide. The peptides were internalized by the bEnd.3 mouse endothelial cells within 30 min of incubation and frequently co-localized with endo-lysosomal vesicles. Moreover, our *in vitro* Transwell translocation experiments confirmed that GYR was able to cross the murine barrier and indicated the successful translocation of MTfp. Thus, despite binding to endocytic receptors with different affinities, both peptides are able to transcytose across the murine BECs.

## Introduction

The brain capillary endothelial cells (BECs) are a key component of the tight blood-brain barrier (BBB), which protects the brain from potentially harmful substances [1–3]. The transport

**Funding:** The Authors has been supported by a grant from the Lundbeck foundation (grant number R155-2013-14113) The funders had no role in study design, data collection and analysis, decision to publish, or preparation of the manuscript.

**Competing interests:** The authors have declared that no competing interests exist.

across the brain endothelium is highly selective. As a result, most small and large molecule brain therapeutics fail to cross the BBB and do not reach their target within the brain parenchyma [1–3]. In order to overcome this hurdle, an endogenous transportation route is often used as a delivery strategy. These routes include carrier-mediated transport, adsorptive- and receptor-mediated transcytosis. Among these strategies, the receptor-mediated transcytosis pathway has been used to transcytose macromolecules, such as monoclonal antibodies, antibody fragments, and peptides, to the brain [4–8]. Peptides are generally easier and less costly to produce and modify than antibodies, making them excellent, specific targeting moieties [9–13]. Therefore, an increasing number of peptides have been investigated for their ability to transcytose from the circulation to the brain [11, 14, 15]. Although a few transcytosing peptides have been identified, details of their target receptor and the transcytotic mechanism are often limited [11, 15–17].

Human melanotransferrin (MTf) is expressed in melanomas as well as in multiple other tissues in lower amounts, including the brain endothelium [18–20]. Interestingly, MTf was shown to cross the BBB to a much greater extent than bovine serum albumin (BSA) and transferrin (Tf) after intravenous injection [21]. Despite the structural similarity between Tf and MTf, competition assays with holo-transferrin indicated that the transferrin receptor (TfR) was not involved in the uptake. Instead, a 50% inhibition of MTf uptake in BECs was obtained using a high concentration of receptor-associated protein, suggesting that the low-density lipoprotein receptor-related protein 1 (LRP-1) is involved in the uptake [21]. Furthermore, when MTf was conjugated to doxorubicin, a chemotherapeutic that does not normally cross the BBB, increased drug accumulation and reduced tumor growth were observed in mice [22]. Interestingly, a peptide derived from MTf showed even higher concentrations in the brain parenchyma than the full-length MTf [23]. Other MTf-derived sequences have also demonstrated transcytotic capabilities *in vitro* [17]. Thus, MTf and MTf-derived peptides are particularly promising as brain delivery agents.

Besides sequences derived from transcytosing proteins, phage display has guided the discovery of novel peptide sequences that can internalize and transcytose BECs. In 2010, van Rooy *et al.* [11] identified seventeen 15-mer peptides that bound to the murine BEC after the infusion of a random peptide library. Two of the sequences had a high affinity to BEC [11, 12]. One of these 15-amino acid peptides were the so-called GYR peptide (GYRPVHNIRGH-WAPG). Recently, it has been demonstrated that the self-assembled GYR core-shell nanoparticles and nanofibers are capable of crossing BBB *in vivo* and that TfR and RAGE act as predominant receptors of this process [24].

This study points to future usages of GYR and MTf-derived peptides for drug delivery to the brain. Therefore, the endocytosis, subcellular trafficking, and translocation of MTf-derived peptide (Mtfp) and GYR were characterized in BEC. We evaluated live uptake into *in vitro*-cultured BECs using spinning disk confocal microscopy. Affinity for transcytosis-relevant receptors on BECs was measured using surface plasmon resonance (SPR).

## Materials and methods

### Peptide synthesis

The peptides were synthesized and labeled with fluorophores on either the N- or C-terminus. One peptide, MTfp (peptide **1**, Table 1), was derived from MTf [17]. The second peptide, GYR (peptide **2**, Table 1), was discovered by the aforementioned phage display [11].

**Materials and instrumentation.** Fmoc-amino acids, HATU, and Fmoc-Sieber-polystyrene resin were purchased from Iris Biotech GMBH (Marktredwitz, Germany), Boc-Gly-OH from Bachem (Bubendorf, Switzerland), TAMRA-NHS from ThermoFisher Invitrogen™ (Oregon,

**Table 1. The peptide sequences MTfp and GYR.**

| Abbreviation | MTfp | GYR |
|---|---|---|
| Peptide no. | 1 | 2 |
| Full sequence | *H*–FRCLVENRGDVPFVTRIR–*NH₂* | *H*–GYRPVHNIRGHWAPGK–*NH₂* |
| Derived from | Portions corresponding to residues 210–229 and 556–575 of the full sequence of melanotransferrin [17] | Discovered by phage display from a library of 15-mer peptides [11, 12] |

USA), and Fmoc-Rink Amide-Tentagel resin (TGRA) from Rapp Polymere (Tübingen, Germany). All other reagents were purchased from Sigma-Aldrich Co. (Merck KGaA, Darmstadt, Germany). Peptides were synthesized by automated solid-phase peptide synthesis (SPPS) using a Biotage® Initiator+ Alstra™ Microwave Peptide Synthesizer. MALDI-TOF spectra were recorded on a Bruker AutoFlex™ MALDI-ToF MS spectrometer, in a positive-ion mode, using 2,4-dihydro-benzoic acid (DHB, 60 mg/mL) spiked with sodium trifluoroacetate in MeCN as a matrix, analyzed using Bruker autoFlex software and reported as m/z in atom mass units. UPLC-MS spectra were recorded on a Waters Acquity UPLC-MS instrument with a Waters Acquity sample manager, Waters Acquity binary solvent manager, equipped with a Waters UPLC XTerra BEH C8 or C18 column (130 Å, 1.7 μm, 2.1x50 mm), using a linear gradient of 5–100% MeCN in water over 6 min, both containing 0.1% formic acid, with a flow rate of 0.4 mL/min, detection at 220–280 nm on a TUV detector and 200–1200 m/z with ES+ and ES- ionization on a QDa single quadrupole detector. The purity of the final products was measured by analytical HPLC on a Shimadzu Nexera-x2 UHPLC instrument, equipped with a Waters XTerra RP C8 or C18 (125 Å, 5 μm, 4.6 x 150 mm) column, and using a linear gradient of MeCN in water, both containing 0.1% TFA, a flow rate of 1 mL/mL, and detection at 220 nm.

**General SPPS procedure.**  The resin was swelled in DMF for 30 min prior to the synthesis. Each coupling reaction was carried out using HATU (3.92 equiv.), collidine (8 equiv.), and the standard Fmoc-protected L-amino acid derivative of residues unless stated otherwise (4 equiv.) in DMF solvent. For all Gly, Tyr, Pro, Val, Asn, Ile, Trp, Ala, and Lys residues, the coupling reactions were heated to 75°C with microwave irradiation for 5 min, while for all Arg, Cys, and His residues, the reactions were instead stirred at room temperature for 30 min. All Leu, Asn, and Val residues and residues after the first ten were coupled twice to ensure full conversion. Deprotection of the $N_\alpha$-Fmoc group was facilitated by two rounds of 20% piperidine in DMF and stirring at room temperature for 3 and 10 min. The resin was washed twice with each DMF and DCM after the synthesis. The peptide identity of the on-resin products was confirmed by suspending a small amount of resin in 100 μL TFA:TIPS:H₂O 95:2,5:2,5 cleavage mixture for 30 min, filtering off the resin, removing the TFA with nitrogen flow and dissolving in water/MeCN (1:1) before confirming the mass by MALDI-TOF and UPLC-MS. Cleavage and full deprotection were achieved by suspending the resin in TFA/TIPS/H₂O/EDT (95:1:2:2) for MTfp or TFA/TIPS/H₂O (95:2.5:2.5) for GYR for 3–4 h with agitation, filtering off the resin, removing the TFA *in vacuo* and precipitation from cold ether.

**Synthesis MTfp (1).**  The resin-bound linear L-octadecapeptide H-Phe-Arg(Pbf)-Cys(Trt)-Leu-Val-Glu(tBu)-Asn(Trt)-Arg(Pbf)-Gly-Asp(tBu)-Val-Pro-Phe-Val-Thr(tBu)-Arg(Pbf)-Ile-Arg(Pbf)-TGRA was synthesized by automated SPPS. It was carried out on a 0.500 mmol scale, starting with 2.51 g of Tentagel Rink Amide resin (TGRA) with 0.25 mmol/g loading. Cleavage of a part of the resin (0.05 mmol) and purification by preparative HPLC (C18 column, 20–50% linear gradient of MeCN in water with 0.1% TFA over 15 min, flow rate 17 mL/min) afforded peptide **1** (37.3 mg, 36%) as a white powder. Purity = 92.3% and $t_r$ = 4.3 min (anal. HPLC, C8 column, 20–80% MeCN in water over 15 min). Calc. mass [M+H]⁺ 2176.2; found mass (MALDI-TOF) [M+H]⁺ 2175.7.

**Synthesis of GYR (2).** The resin-bound linear L-pentadecapeptide Boc-Gly-Tyr(tBu)-Arg (Pbf)-Pro-Val-His(Trt)-Asn(Trt)-Ile-Arg(Pbf)-Gly-His(Trt)-Trp(Boc)-Ala-Pro-Gly-Lys(Mtt)-SR was synthesized by SPPS. It was carried out on a 0.100 mmol scale, starting with 165.7 mg of Sieber resin (SR) with 0.61 mmol/g loading. Fmoc-Lys(Mtt)-OH and Boc-Gly-OH were used instead of standard Fmoc-amino acids for the first and last coupling, respectively. Cleavage and purification by preparative HPLC (C18 column, 20–80% linear gradient of MeCN in water with 0.1% TFA over 30 min, flow rate 17 mL/min) afforded peptide **2** (38.2 mg, 20%) as a white powder. Purity = 95.8% and $t_r$ = 5.0 min. (UPLC, C8 column, 5–60% MeCN in water over 6 min). Calc. mass [M+H]$^+$ 1844.0; found mass (MALDI-TOF) [M+H]$^+$ 1844.9.

**Synthesis of TAMRA-labeled MTfp (3).** To the pre-swelled resin-bound side-chain protected MTfp (**1**) (0.01 mmol) was added TAMRA-NHS (38.9 mg, 0.100 mmol, 2.0 equiv.) in dry DMF, followed by triethylamine (30 μL, 0.409 mmol, 8.2 equiv.). The mixture was agitated overnight at room temperature under nitrogen atmosphere and protected from light. The resin was washed three times with each DMF and DCM. Cleavage and purification by preparative HPLC (C18, 5–50% linear gradient of MeCN in water with 0.1% TFA over 30 min, flow rate 20 mL/min) afforded TAMRA-labeled peptide **3** (4.7 mg, 18%) as a pink powder. Purity = 99.7% and $t_r$ = 8.8 min (anal. HPLC, C8 column, 5–100% MeCN in water over 15 min). Calc. mass [M+H]$^+$ 2588.3; found mass (MALDI-TOF) [M+H]$^+$ 2587.8.

**Synthesis of TAMRA-labeled GYR (4).** The resin-bound linear L-pentadecapeptide Boc-Gly-Tyr(tBu)-Arg(Pbf)-Pro-Val-His(Trt)-Asn(Trt)-Ile-Arg(Pbf)-Gly-His(Trt)-Trp(Boc)-Ala-Pro-Gly-Lys(Aloc)-TGRA was synthesized by automated SPPS. It was carried out on a 0.250 mmol scale, starting with 1.14 g of Tentagel Rink Amide resin (TGRA) with 0.25 mmol/g loading. Fmoc-Lys(Alloc)-OH and Boc-Gly-OH were used instead of standard Fmoc-amino acids for the first and last coupling, respectively. The Alloc protection was removed by adding 5 mL dry DCM, phenylsilane (406 μL, 13 equiv.) and tetrakis(triphenylphosphine)palladium(0) (30 mg, 0.1 equiv.) in 1 mL DCM to the pre-swelled resin under nitrogen atmosphere and agitating for 15 min. The procedure was repeated four times to ensure full removal of the Alloc protection group before washing the resin five times with DCM. TAMRA-NHS (10.1 mg, 1 equiv.) was added to part of the pre-swelled resin (0.029 mmol) in dry DMF/DCM (1:1, 1 mL) followed by triethylamine (4 μL, 3 equiv.). The mixture was agitated overnight at room temperature under nitrogen atmosphere and protected from light. The resin was washed three times with each DMF and DCM. Cleavage and purification by preparative HPLC (C18 column, 5–50% linear gradient of MeCN in water with 0.1% TFA over 30 min, flow rate 20 mL/min) afforded TAMRA-labeled peptide **4** as a pink powder (28.1 mg, 43%). Purity = 100% and $t_r$ = 7.3 min. (anal. HPLC, C8 column, 5–100% MeCN in water over 15 min). Calc. mass [M+H]$^+$ 2256.1; found mass (MALDI-TOF) [M+H]$^+$ 2255.8.

## Surface Plasmon Resonance (SPR)

The binding affinity analysis of the MTf and GYR peptides was performed using the Biacore 3000 system (Cytiva, UK) equipped with a CM5 sensor chip. The sensor chip was initially activated by injection of 0.2 M 1-ethyl-3-(3-dimethylaminopropyl) carbodiimide hydrochloride and 0.05 M N-hydroxysuccinimide in water. Recombinant human and mouse TfR protein (SinoBiological, Inc., Beijing, PRC) was purchased with a His-tag and immobilized on the sensor chip at 58 fmol/mm$^2$. LRP-1 was purified from human placenta according to a previously described protocol [25] and immobilized to densities of 0.025 fmol/mm$^2$. The remaining carboxylate groups were blocked with 1 M ethanolamine. The MTf and GYR peptides were injected for binding to the immobilized TfR or LRP-1 protein at 5 μL/min at 25°C in 10 mM HEPES, 150 mM NaCl, 1.5 mM CaCl$_2$, 1 mM EGTA, and 0.005% Tween 20 (pH 7.4) at five

different concentrations (0.25 μM, 0.5 μM, 1 μM, 2 μM and 4 μM). The reference flow channel (FC1) was activated and blocked but without immobilized TfR or LRP-1. The dissociation constant ($K_D$) values were calculated using the BIAevaluation 4.1 software (Cytiva, UK) using the predefined Langmuir 1:1 interaction model and fitted using a drifting baseline with global fitting to the curves of the considered concentration range.

## Routine cell culturing

Immortalized mouse BEC, bEnd.3 (ATCC, Cat no. CRL-2299) between passage 30 and 38 were used. For routine cell culture, the cells were seeded in rat-tail type I collagen (100 μg/ml, Sigma, Cat no. C3867-1VL) pre-coated flasks (Thermo Scientific™ Nunc™ Cell Culture Treated EasYFlasks™ or Greiner Bio-One GmbH) and were cultured in Dulbecco's Modified Eagle Medium (DMEM) (Sigma, Cat no. D0819) supplemented with 10% fetal bovine serum (FBS) (Sigma, Cat no. F9665), 1% penicillin-streptomycin (Thermo Fisher Scientific, Cat no. 15140–122), and 1 mM sodium pyruvate (Sigma, Cat no. S8636); referred as cDMEM. Please note that all cell cultureware was coated with 100 μg/mL rat-tail type I collagen (Sigma, Cat no. C3867-1VL) diluted in $ddH_20$ for 1 hour @ 37˚C prior to cell seeding. Cells were cultured in an incubator at 37˚C with 5% $CO_2$/95% air and saturated humidity. The cell culture medium was changed every three days. Cells were sub-cultured once they reached 85–90% confluency.

## Immunocytochemistry

The bEnd.3 cells were seeded on collagen-coated 8-well imaging chamber (10,000 cells/well) (Thermo Fisher Scientific, Lab-Tek® II Chamber Slide™ Cat no. 155409) or polyester Transwell inserts (40,000 cells/insert) (Corning Cat no. 3460) and grown for 5 days, until they formed a confluent monolayer. On the day of the experiment, the cells were washed with preheated cDMEM before fixation with 4% paraformaldehyde (Sigma, Cat no. 441244) in phosphate buffer saline (PBS) for 10 min at room temperature, followed by three PBS washes. The cells were permeabilized with 0.2% Triton X-100 (Sigma, Cat no. X100) (PBS-TX) in PBS for 10 min and blocked in 2% BSA (VWR, Cat no. 0332) (in 0.05% PBS-TX) for 20 min. For the LRP-1 and TfR surface staining, the cells were not permeabilized, and the BSA was diluted in PBS only. Cells were then incubated with primary antibody diluted in 2% BSA for 1 hour at room temperature (see list of antibodies in Table 2).

After incubation, cells were washed three times for 5 min with PBS and incubated with fluorescently labeled secondary antibodies (Table 2) for 1 hour in dark at room temperature, followed by rinsing three times 5 min with PBS. Then, the cells were quickly washed

**Table 2. List of antibodies.**

| Antigen | Antibody type | Manufacturer, catalog number | Concentration |
|---|---|---|---|
| Claudin-5 | Mouse monoclonal, Clone 4C3C2 | Thermo Fisher Scientific, 35–2500 | 2.5 μg/mL |
| CD31 (PECAM-1) | Mouse monoclonal, Clone 2H8 | DSHB, 2H8 | 1.0 μg/mL |
| ZO-1 | Mouse monoclonal | Thermo Fisher Scientific, 61–7300 | 2.5 μg/mL |
| LRP-1 | Rabbit polyclonal | N/A | 1.5 μg/mL |
| TfR | Mouse monoclonal with human FC, Clone 8D3 | Provided by H. Lundbeck A/S | 6 μg/mL |
| Mouse IgG (H+L) | Goat polyclonal, Alexa Fluor 488-coupled | Thermo Fisher Scientific, A-11001 | 2 μg/mL |
| Rabbit IgG (H+L) | Goat polyclonal, Alexa Fluor 488-coupled | Thermo Fisher Scientific, A-11008 | 2 μg/mL |
| Rabbit IgG (H+L) | Goat polyclonal, Alexa Fluor 647-coupled | Thermo Fisher Scientific, A-21244 | 2 μg/mL |
| Human IgG (H+L) | Goat polyclonal, Alexa Fluor 488-coupled | Thermo Fisher Scientific, A-11013 | 2 μg/mL |

twice with ddH$_2$O and the nuclei of bEnd.3 cells were stained with Hoechst 33342 (1 μg/ml concentration in ddH$_2$O) (Sigma, Cat no. B2261) for 15 min in dark at room temperature. Finally, the cells were washed twice for 2 min with ddH2O and were kept in PBS until imaging. The cells were observed and photographed using a spinning disk confocal microscopy system consisting of a CSU-X1 spinning disk unit (Yokogawa Electric Corporation, Japan) and an Andor iXon-Ultra 897 EMCCD camera (Andor, UK), mounted on an inverted fully motorized Olympus IX83 microscope body and a UPlanSApo 60x/NA1.20 (WD = 0.28 mm) water immersion objective (Olympus Corporation, Japan). The following excitation laser lines and emission filters were used: tight junction proteins (Alexa Fluor 488-coupled secondary antibody): $\lambda_{ex}$ = 488 nm, $\lambda_{em}$ = 525/50 nm bandpass filter and nuclei (Hoechst 33342): $\lambda_{ex}$ = 405 nm, $\lambda_{em}$ = 440/521/607/700 nm quad-band bandpass filter. The raw images were processed using Imaris (version 8.2.1, Bitplane AG, Switzerland).

## Subcellular localization studies of MTf and GYR peptides

One hundred twenty-five thousand bEnd.3 cells were seeded on collagen-coated 35 mm glass-bottom dishes (MatTek Cat no. P35G-1.5-14-C) and were grown for 4–5 days in cDMEM, until they formed a confluent monolayer. On the day of the experiment, bEnd.3 cells were incubated with 1 mL of 10 μM TAMRA-labeled MTf and GYR peptides in phenol red-free cDMEM for 30 min. After the 30 min exposure time, the medium was removed, and the samples were washed three times with peptide-free cDMEM, and finally, fresh medium was added to the cells. The imaging chambers were kept at 37˚C with 5% CO$_2$/95% air and saturated humidity during imaging. Following peptide incubation and prior to image acquisition, the acidic organelles were stained with 200 nM LysoTracker™ Green DND-26 (Thermo Fisher Scientific, Cat no. L7526) for 3 min, and the cell membrane was labeled with 5 μg/ml Wheat Germ Agglutinin, Alexa Fluor™ 488 Conjugate (WGA) (Thermo Fisher Scientific, Cat no. W11261) for 5 min and washed once with cell culture medium. Three-dimensional images ('z-stacks') were obtained using the spinning disk confocal microscopy system described above with a UPlanSApo 60x/NA1.20 (WD = 0.28 mm) water immersion or a UPlanSApo 100xS/NA1.35 (WD = 0.20 mm) silicone immersion objectives (Olympus Corporation, Japan). The following excitation laser lines and emission filters were used: WGA, LysoTracker™ Green: $\lambda_{ex}$ = 488 nm, $\lambda_{em}$ = 525/50 nm bandpass filter, TAMRA-labeled MTf and GYR peptides: $\lambda_{ex}$ = 561 nm, $\lambda_{em}$ = 625/90 nm bandpass filter. Images were acquired using Olympus cellSens software (version 1.18) and processed using Imaris imaging software (version 8.2.1).

## Energy-dependent uptake of MTf and GYR peptides

One hundred twenty-five thousand bEnd.3 cells were seeded on collagen-coated 35 mm glass-bottom dishes (MatTek Cat no. P35G-1.5-14-C) and were grown for 5 days in cDMEM, until they formed a confluent monolayer. On the day of the experiment, bEnd.3 cells were labeled with 5 μg/ml Alexa Fluor™ 488 conjugated WGA (Thermo Fisher Scientific, Cat no. W11261) for 5 min and washed once with cell culture medium. Then ice-cold phenol red-free cDMEM was added to the cells, and the cells were pre-incubated at 4˚C for 15 min prior to peptide exposure. The bEnd.3 cells were incubated with 1 mL of 10 μM TAMRA-labeled MTf and GYR peptides in phenol red-free cDMEM at 4˚C for 1 hour. Then the peptide solution was removed, and the cells were washed once with ice-cold PBS containing heparin, and warm (37˚C) medium was added to the cells. Immediately after that, three-dimensional images were obtained using the spinning disk confocal microscopy system described above with a UPlanSApo 100xS/NA1.35 (WD = 0.20 mm) silicone immersion objectives (Olympus Corporation,

Japan) and climate control chamber (37°C, controlled $CO_2$ and humidity), *i.e.*, 0 min chase. The three-dimensional images were obtained from several regions every 30 min for 2 hours. The following excitation laser lines and emission filters were used: WGA, LysoTracker™ Green: $\lambda_{ex}$ = 488 nm, $\lambda_{em}$ = 525/50 nm bandpass filter, TAMRA-labeled MTf and GYR peptides: $\lambda_{ex}$ = 561 nm, $\lambda_{em}$ = 625/90 nm bandpass filter. Images were acquired using Olympus cellSens software (version 1.18) and processed using Arivis Vision 4D (version 3.3.0).

## Co-localization analysis

Lysosomal co-localization was determined using Imaris XTension. First, all peptides and lysosomes were identified as spots using the built-in 'spot detection' algorithm. The 'quality intensity threshold' parameters were adjusted manually for each image to account for differences in background intensity. The threshold was adjusted until the majority of identifiable peptides and lysosomes were labeled. Spots located within 1 μm distance were identified as co-localized spots. The Mander's overlap coefficient (MOC) was calculated based on the following formula:

$$MOC = \frac{No.\ of\ peptide\ spots\ co-localised\ with\ lysosomes}{All\ identified\ spots}.$$

The final MOC represents the mean and standard deviation (SD) of three independent experiments; 10–14 regions were imaged in each experiment.

## Translocation studies of MTf and GYR peptides

For the translocation studies, an *in vitro* contact and non-contact co-culture model was used. Forty thousand bEnd.3 cells were seeded on the upper side of the collagen-coated Transwell insert and maintained in cDMEM for 24 hours (contact co-culture) or immediately transferred into a 12-well plate in which rat astrocytes were grown for at least three weeks (non-contact co-culture). In the case of contact co-culture, 24 hours after plating the bEnd.3 approx. 150,000 rat astrocytes were seeded on the bottom side of the flipped Transwell inserts, which was coated with 5 μg/ml poly-L-lysine (P1524, Sigma-Aldrich), and incubated for 2 hours at 37°C. The inserts were then flipped back to the original side and placed into a 12-well plate with rat astrocytes. The cells were cultured in cDMEM at 37°C with 5% $CO_2$/95% air and saturated humidity, and the experiments were performed 4.5–5 days post-seeding of bEnd.3 cells. The medium was switched to phenol red-free cDMEM the day before the experiment.

**Translocation study using live-cell imaging.** These experiments were performed using the contact co-culture model. On the day of the experiment, the inserts were placed into an empty 12-well plate, and the cells were gently washed with phenol red-free cDMEM, followed by a medium exchange on both the apical and basolateral sides. The apical side of the membrane was exposed to 10 μM MTfp or GYR for 30 min, then washed once with phenol red-free cDMEM, and both sides of the membrane were stained with 5 μg/ml Alexa Fluor™ 488 conjugated WGA (Thermo Fisher Scientific, Cat no. W11261) for 3 min. Both sides of the membrane were gently washed with phenol red-free cDMEM, and the inserts were placed into a glass-bottom imaging dish filled with experimental medium. The cells were imaged with the same spinning disk microscopy system and settings as described previously. Images were processed using Arivis Vision 4D (version 3.3.0).

**Translocation study based on fluorescence intensity measurement.** These experiments were performed using the bEnd.3 cells that were cultured with rat astrocytes in a non-contact co-culture model and with cell-free filters as negative controls. On the day of the experiment, the inserts were transferred to a new astrocyte-free 12-well plate, and the medium was changed on both the apical and basolateral sides. The apical side of the membrane was exposed to 10 μM

MTfp or GYR for 2 hours while the plates were gently shaken to overcome unstirred water-layer effects. After the 2 hours exposure time, 150 μL sample was taken from both apical and basolateral compartments, and the fluorescence intensity was read by a CLARIOstar® Plus microplate reader (BMG Labtech) ($\lambda_{Exc}$ = 535 nm, $\lambda_{Em}$ = 585 nm, Bandwidth = 20 and 30 nm, respectively). The translocation of the peptides is presented as permeability coefficient (a concentration independent transport parameter) and was determine by using the clearance principle [26].

The following equations [27] were used to calculate the permeability coefficient:

$$P_{app}\left[\frac{cm}{s}\right] = \frac{B}{T} \cdot \frac{v_b}{A \cdot t \cdot 60} \tag{1}$$

Where $P_{app}$ is the apparent permeability, B is the relative fluorescence unit (RFU) at time t, T is the top chamber RFU at time 0 (constant), $V_b$ is the volume of the bottom channel [ml], A is the cross-section area of the membrane [cm$^2$], and t is the time [min].

The final endothelial permeability ($P_e$) was calculated based Eq (2):

$$\frac{1}{P_e} = \frac{1}{P_{app, \ bEnd.3+filter}} - \frac{1}{P_{app, \ filter}} \tag{2}$$

Where $P_{app, \ bEnd.3+filter}$ is the apparent permeability of the bEnd.3 model (cells + filter), and $P_{app, \ filter}$ is the apparent permeability of the collagen-coated blank porous membrane.

The translocation experiments were performed three times in technical triplicates, and the data are represented as mean (SD).

## Paracellular permeability of 4 kDa dextran

The paracellular permeability of fluorescein isothiocyanate (FITC) labeled 4 kDa dextran (FD4, Sigma Aldrich) to determine the tightness of our bEnd.3 model.

The bEnd.3 cells were cultured with rat astrocytes as described above. On the day of the experiment, the inserts were transferred to a new astrocyte-free plate, and the medium was changed on both the apical and basolateral sides. To initiate the permeability experiment and avoid the temporary disruption of the barrier, 50 μL medium was removed from the apical side and 50 μL 200 μM working solution of 4kDa FITC-dextran was added to the upper well, *i. e.* the apical well had 500 μl 10 μM 4kDa FITC-dextran. In every 15 min for a period of 60 min, starting from 0 min, 100 μl medium was removed from the basolateral side. At the final time point, 100 μl medium was also removed from the apical compartment. The fluorescence intensity of the sample was read by a CLARIOstar® Plus microplate reader (BMG Labtech) ($\lambda_{Exc}$ = 485 nm, $\lambda_{Em}$ = 515 nm, Bandwidth = 15 and 20 nm, respectively), and the paracellular permeability was calculated according to Eqs (1) and (2). The permeability experiments were repeated three times in technical triplicates, and the data are represented as mean (SD).

## Statistical analysis

The normality of the co-localisation, permeability, and translocation data was tested and confirmed with GraphPad Prism 9.0.2 (GraphPad Software, Inc, CA, USA) using its built-in normality tests and QQ plots (not shown). Since the experimental data exhibited a normal/Gaussian distribution, they are presented as mean (SD).

## Results

### Peptide synthesis and fluorophore labeling

We synthesized two peptides (MTfp and GYR) by standard Fmoc SPPS, followed by cleavage and purification by preparative HPLC. They were used for SPR (peptide 1 and 2, respectively) to evaluate the binding affinities to key receptors. To study their uptake *in vitro*, the peptides were labeled with the tetramethylrhodamine (TAMRA) fluorophore. The MTfp was labeled on-resin at the free N-terminal amine with an amine-reactive NHS ester of the TAMRA carboxylic acid derivative (TAMRA-NHS) to form an amide bond. The peptide with the N-terminal TAMRA (**3**) was subsequently cleaved off the resin and purified by preparative HPLC. The synthesis and TAMRA labeling of MTfp is outlined in Fig 1. In order to conjugate a fluorophore to the C-terminus of GYR on-resin, a lysine residue with an orthogonal $N_\epsilon$-protection group was added to the C-terminus of the peptide sequence. The Alloc-protection group was removed by the palladium-catalyzed transfer to a scavenger such as phenylsilane [28, 29] and was, therefore, fully orthogonal to the acid- and base-labile protection groups used in the standard Fmoc peptide synthesis. The GYR peptide was synthesized by SPPS, the Alloc protection removed, and the free amine on the C-terminal lysine reacted with TAMRA-NHS. Full cleavage and purification by preparative HPLC afforded the TAMRA-labeled GYR (**4**). The synthesis and TAMRA labeling of GYR is outlined in Fig 2. The peptides were afforded in moderate yields and measured in purities above 90% by HPLC.

### Binding affinity to TfR and LRP-1

TfR and LRP-1 expression in bEnd.3 cells were confirmed by Western blotting (not shown). By using SPR, we measured the binding affinities and kinetics of the MTfp and GYR (peptide **1** and **2**, respectively) to endothelial cell surface receptors: human and mouse TfRs, and human LRP-1. The equilibrium dissociation constants ($K_D$) were determined based on the SPR sensorgrams (Fig 3). For all three immobilized receptors, the $K_D$ values of MTfp were several magnitudes lower than those of GYR. The binding affinities for both MTfp and GYR decreased in the following order: LRP-1 > human TfR > mouse TfR.

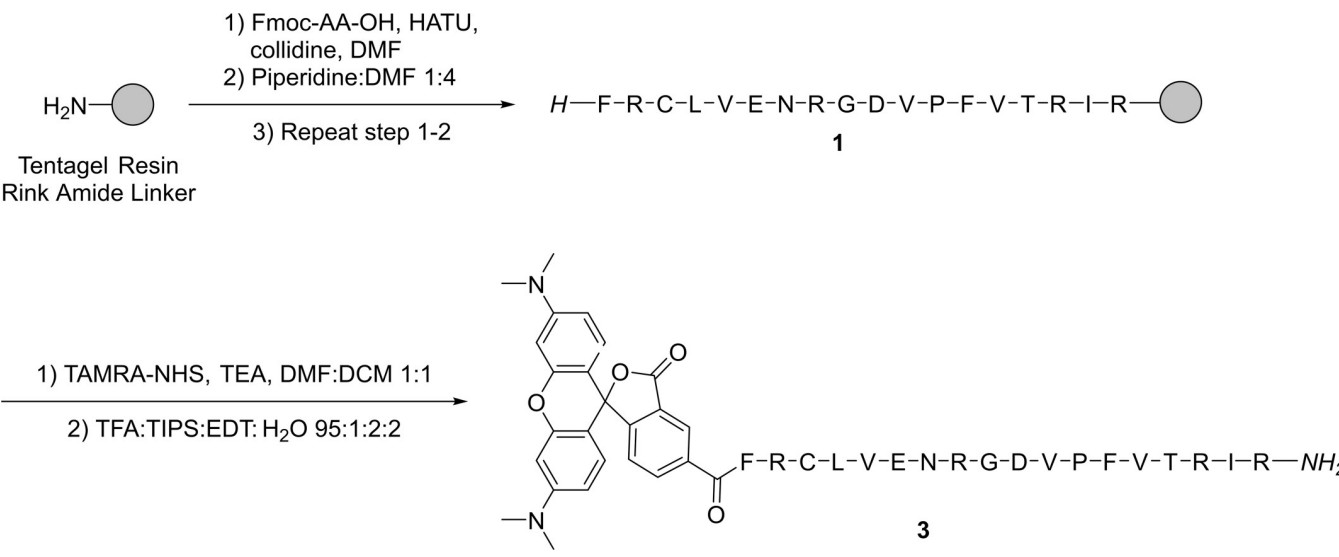

**Fig 1. The synthesis of unlabeled (1) and TAMRA-labeled MTfp (3).** Peptide sequences are shown using the one-letter code.

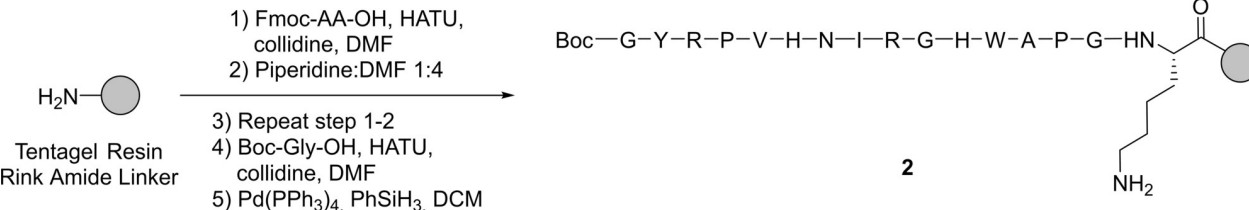

**Fig 2. The synthesis of unlabeled (2) and TAMRA-labeled GYR (4).** Peptide sequences are shown using the one-letter code.

## Cellular uptake of MTfp and GYR

Next, we investigated the cellular uptake and localization of TAMRA-labeled MTfp and GYR (peptides **3** and **4**, respectively) in confluent bEnd.3 cells. Since the cells were grown on glass-

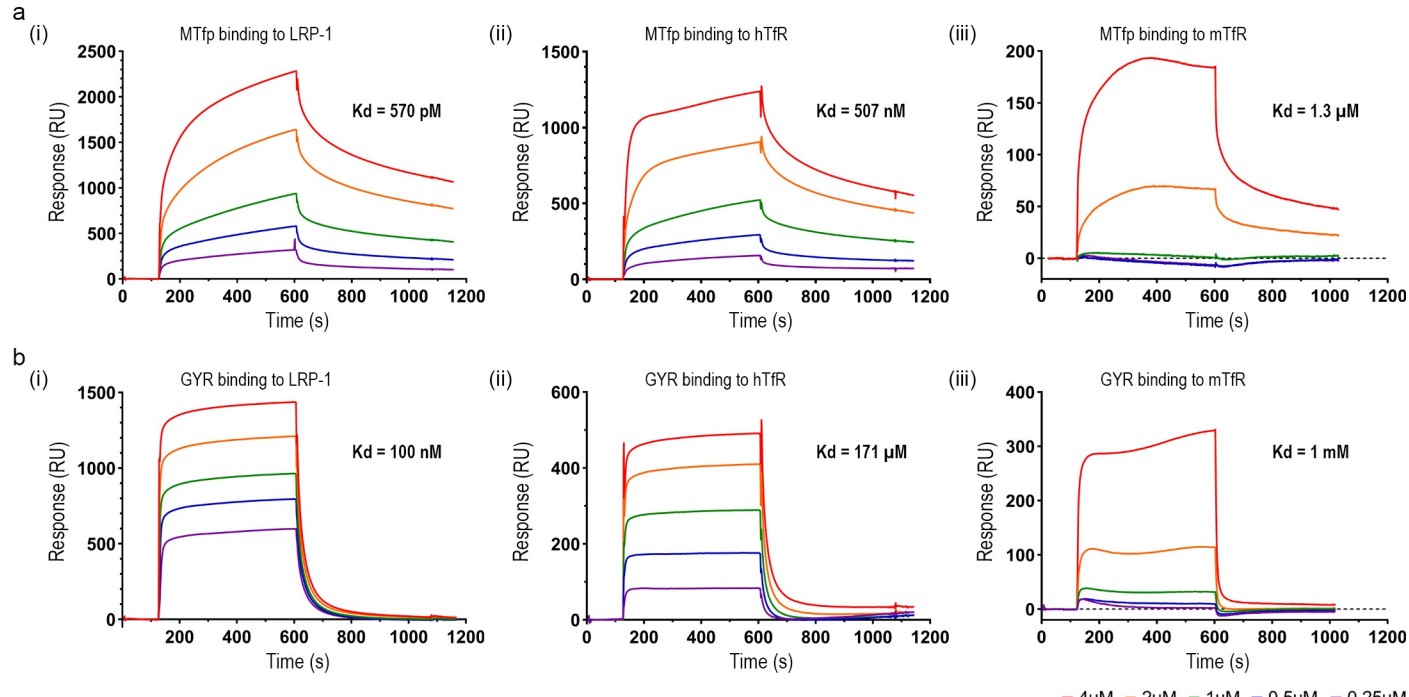

**Fig 3. Receptor binding.** Binding affinities of MTfp **1** (a) and GYR **2** (b) to LRP-1 (i), human TfR (hTfR) (ii), and mouse TfR (mTfR) (iii). $K_D$ is denoted for each equilibrium.

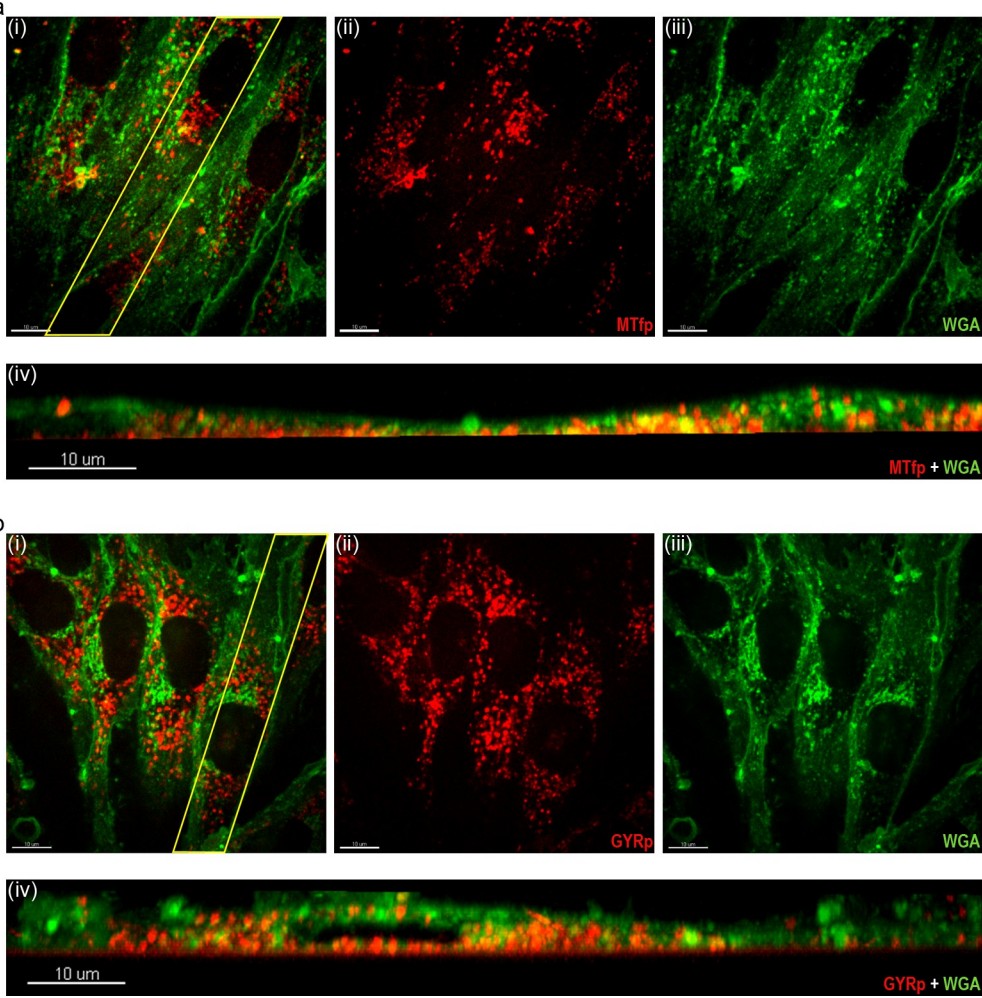

**Fig 4. Internalization of TAMRA-labeled MTfp and GYR.** Uptake of 10 μM TAMRA-labeled MTfp 3 (a) and GYR 4 (b) in confluent bEnd.3 cells 15 min after a 30-min peptide exposure. (i-iii) 2D micrographs of the middle section of the WGA labeled bEnd.3 cells (green). (iv) Cross-sectional view of the randomly selected 15 μm thick region (indicated with yellow in subfigure a(i) and b(i)). Scale bars: 10 μm, image acquisition: 100x silicone immersion objective.

bottom imaging dishes, the endothelial cell barrier's integrity and tightness were validated by the expression of endothelial junctional proteins using immunocytochemistry. S1 Fig shows that bEnd.3 cells expressed claudin-5, CD31, and ZO-1 junctional proteins. Live-cell imaging showed that both MTfp and GYR (red) were taken up after a 30-minute incubation (representative images in Fig 4). Both peptides were detected in all horizontal planes of the BECS, also on the lower (basal) cell membrane indicating translocation (cross-sectional view, Fig 4A(iv) and 4B(iv)). This uptake was energy-dependent and most likely receptor-mediated, as we have not seen any significant peptide uptake even after 2 hours (37˚C) following a 1-hour peptide exposure at 4˚C (S2 and S3 Figs). Furthermore, surface staining of LRP-1 and TfR demonstrated that LRP-1 is primarily located on the basolateral membrane, whereas TfR seems to be located on the apical side as well (S4 Fig). This indicates that TfR is a more important receptor for the observed uptake.

Moreover, live-cell imaging suggests that both MTfp and GYR accumulate in intracellular vesicles (Fig 5A(i–iii) and 5B(i–iii)). To further investigate the peptides' subcellular

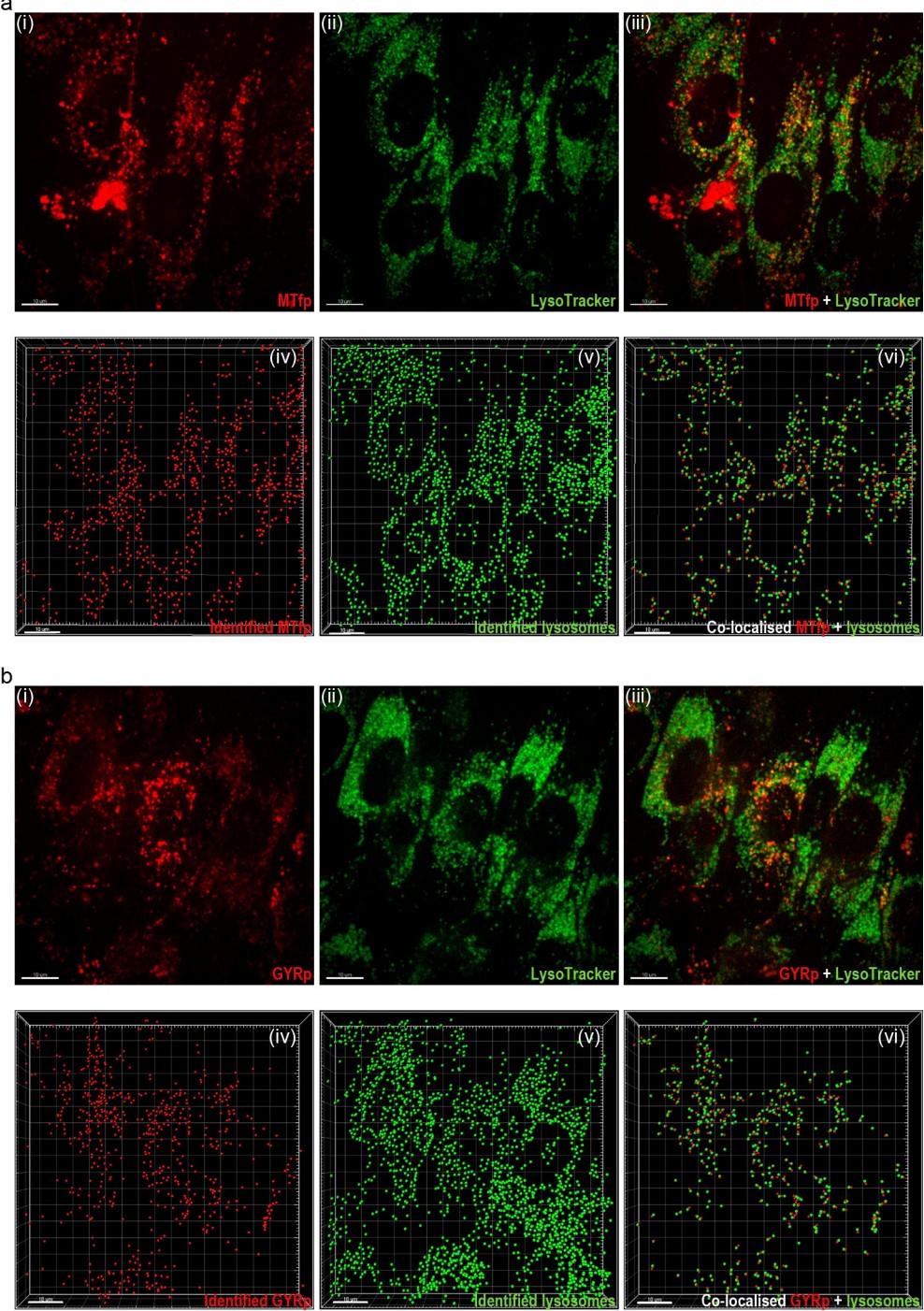

**Fig 5. Cellular endocytosis and localization.** Subcellular localization of 10 μM TAMRA-labeled MTfp 3 (a) and GYR 4 (b) in confluent bEnd.3 cells 15 min after a 30-minute peptide exposure. (i–iii) 2D micrographs of the middle section of the LysoTracker™ Green labeled bEnd.3 cells (lysosomal stain, green). (iv–v) 3D rendering showing all identified peptides and lysosomes as spots and in (vi) the co-localized spots. Green; lysosomes, red: MTfp and GYR. Scale bars: 10 μm, image acquisition: 100x silicone immersion objective.

localization and determine whether they accumulated in acidic organelles (mainly lysosomes), bEND.3 cells were labeled with LysoTracker™ Green. The raw data suggested high lysosomal

accumulation of both MTfp and GYR (representative images in Fig 5A(i–iii) and 5B(i–iii)). Lysosomal co-localization was determined by first identifying the peptides and lysosomes as individual spots using the built-in 'spot detection' algorithm (Fig 5A(iv–v) and 5B(iv–v)). Spots located within 1 μm distance (default spot co-localization setting of Imaris XTension) were identified as co-localized spots (Fig 5A(vi) and 5B(vi)). Mander's coefficients of MTfp and GYR co-localization with lysosomes were 0.57 (0.07) and 0.55 (0.06), respectively, suggesting that a substantial fraction of the endocytosed peptides followed the endo-lysosomal pathway.

### Translocation of MTfp and GYR peptides

To further investigate the peptides' BEC crossing ability, we set-up an *in vitro* contact and non-contact co-culture model, in which the bEnd.3 cells were co-cultured with rat astrocytes (rAstro). The integrity of the barrier was confirmed by immunostaining for tight junction markers (not included) and permeability measurement of 4 kDa FITC-Dextran (Table 3). The translocation was assessed by live-cell imaging and fluorescence intensity-based studies. As Figs 6 and 7 show, both MTfp and GYR peptides crossed the bEnd.3 monolayer after 30-min peptide exposure) however, most of them were trapped on the basal side of the membrane pores, and MTfp and GYR peptides were visually not found inside the rAstro. Fig 6B and 7B (cross-sectional view of the contact co-culture model) clearly show that most of the peptides were located on the basolateral side of the membrane (the same was also observed with cell-free controls, see S5 Fig). The translocation was quantified by measuring the fluorescence intensity of the apical and basolateral compartment following a 2-hour peptide uptake and calculating the permeability coefficient. The apparent ($P_{app}$) and "real endothelial" permeability ($P_e$) values are presented in Table 3. GYR had approx. 3 times higher permeability coefficient than MTfp, indicating a better translocation ability through the BBB. However, our $P_{app}$ results suggest MTfp is bound to the collagen-coated polyester membrane or trapped inside the pores of the membrane as the $P_{app}$ of 4kDa Dextran across empty filter was 1.45 times higher than the $P_{app}$ of MTfp (approx. 1.5 less molecular weight; i.e. smaller size and consequently higher expected permeability).

### Discussion

Transport across the BBB and targeting brain diseases is difficult, and thus mechanistic insight into how transport is governed could lead to more efficient therapies. Peptides have shown promise as targeting ligands [14, 23, 30, 31], and here we show that two BBB-targeting peptides binding to relevant receptors are taken up and transported in cultured BECs. MTfp and GYR both show strong binding to LRP-1 and a less favorable binding to TfR.

Using solid-phase synthesis, we conjugated the photostable TAMRA fluorophore at either the N- or C-terminus of MTfp and GYR, respectively. By doing so, we could track the live uptake of these relevant brain-targeting peptides *in vitro*. Previously only quantitative assays

**Table 3. Paracellular permeability and peptide translocation studies.** Permeability coefficients of 4 kDa FITC-Dextran, TAMRA-labeled MTfp and GYR. Permeability (P) is presented as mean (SD) of three independent experiments with triplicates.

| Name | MW (Da) | $P_{app,F}$ (x $10^{-6}$ cm/s) | $P_{app, bEnd.3+F}$ (x $10^{-6}$ cm/s) | $P_e$ (x $10^{-6}$ cm/s) |
|---|---|---|---|---|
| FITC-Dextran | ~4000 | 8.49 (1.90) | 3.32 (0.48) | 5.79 (0.44) |
| MTfp-TAMRA | 2589.0 | 5.86 (1.27) | 2.55 (0.45) | 4.57 (0.64) |
| GYR-TAMRA | 2256.6 | 12.82 (3.11) | 6.64 (1.16) | 15.17 (1.83) |

MW- molecular weight, $P_{app,F}$−apparent permeability through filter, $P_{app,bEnd.3+F}$−apparent permeability through bEnd.3 grown on filter, $P_e$−"real" endothelial permeability. Membrane specifications: Polyester (PET) membrane with 10 μm thickness, 0.4 μm pore size, with $4 \times 10^6$ pores/cm$^2$, *i.e.* 0.50% porosity (Corning #3460)

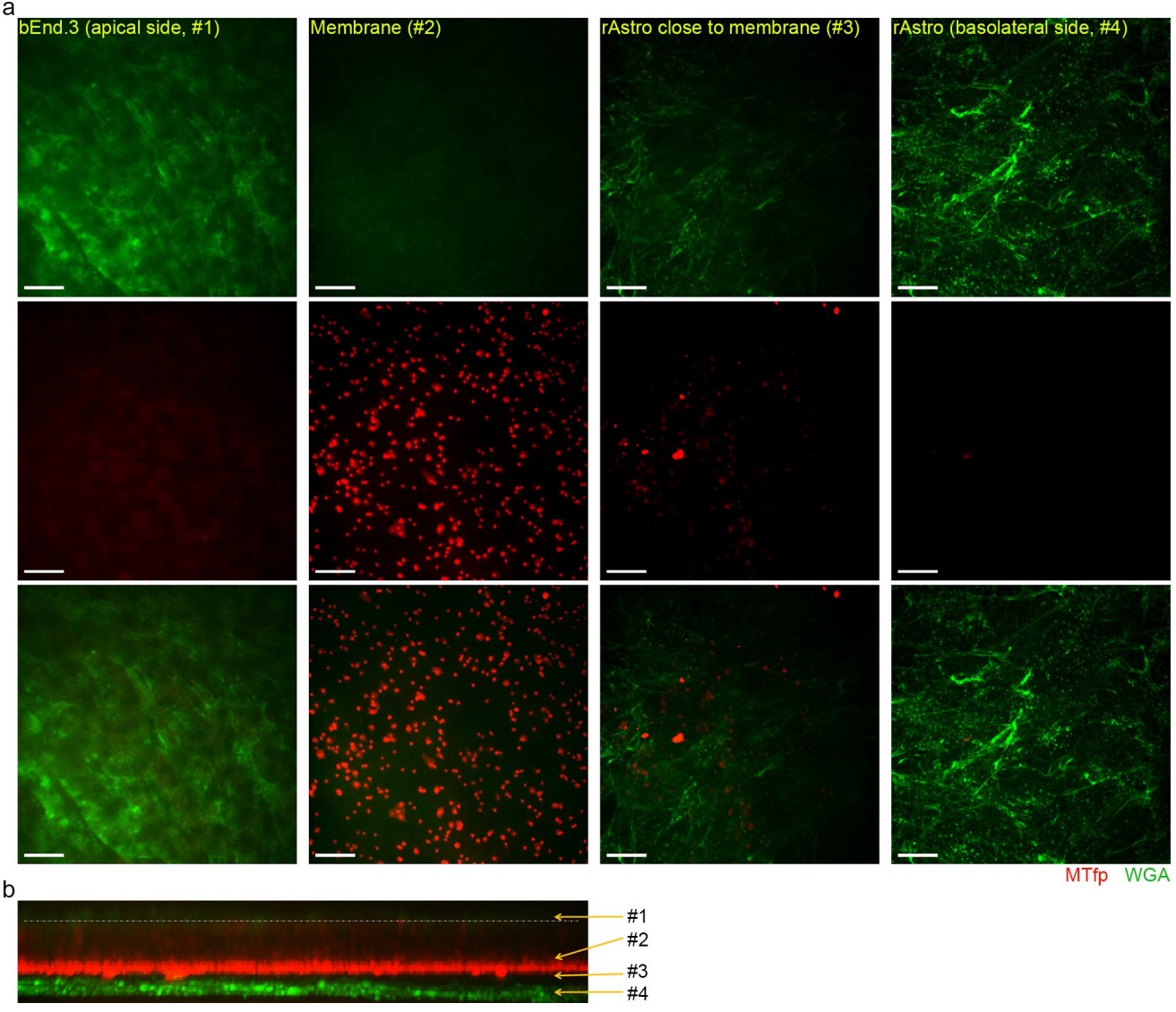

**Fig 6. Translocation of MTfp (3).** Translocation of 10 μM TAMRA-labeled MTfp across confluent bEnd.3 cells cultured as contact co-culture with rAsto (a-b) or monoculture (c). (a) Representative 2D micrographs of different regions of the co-culture show that most of the MTfp is trapped inside the membrane pores (red: MTfp, green: WGA labeled cells). Scale bars: 20 μm. (b) Cross-sectional view of the entire field-of-view. The white dashed line marks the top part of the 10 μm thick membrane. Image acquisition: 60x water immersion objective.

have documented their improved BBB translocation [11, 12], but fluorescence microscopy enables studies on uptake dynamics as well as intracellular localization and sorting [7, 32]. One observation made was that a significant fraction of the peptide was co-localized with the endo-lysosomal system. Ideally, the peptide shuttle should stimulate the fusion of endosomes to tubules facilitating translocation [33], yet in a monocellular system, translocation is not a relevant output. The 4 kDa FITC-Dextran was used to measure the paracellular permeability of the model and to determine whether our observed peptide permeability was due to the leakiness of the modell or not. The literature is lacking of 4 kDa Dextran permeability data on bEnd.3 cells and only one paper was found, the measured permeability coefficient was ranging from 2.48 to 4.01 x $10^{-6}$ cm/s with a mean $P_e$ of 2.91 (0.43) x $10^{-6}$ cm/s [34]. Our model was a bit leakier than this, as we measured a $P_e$ of 5.79 (0.44) x $10^{-6}$ cm/s; this permeability value is similar to the

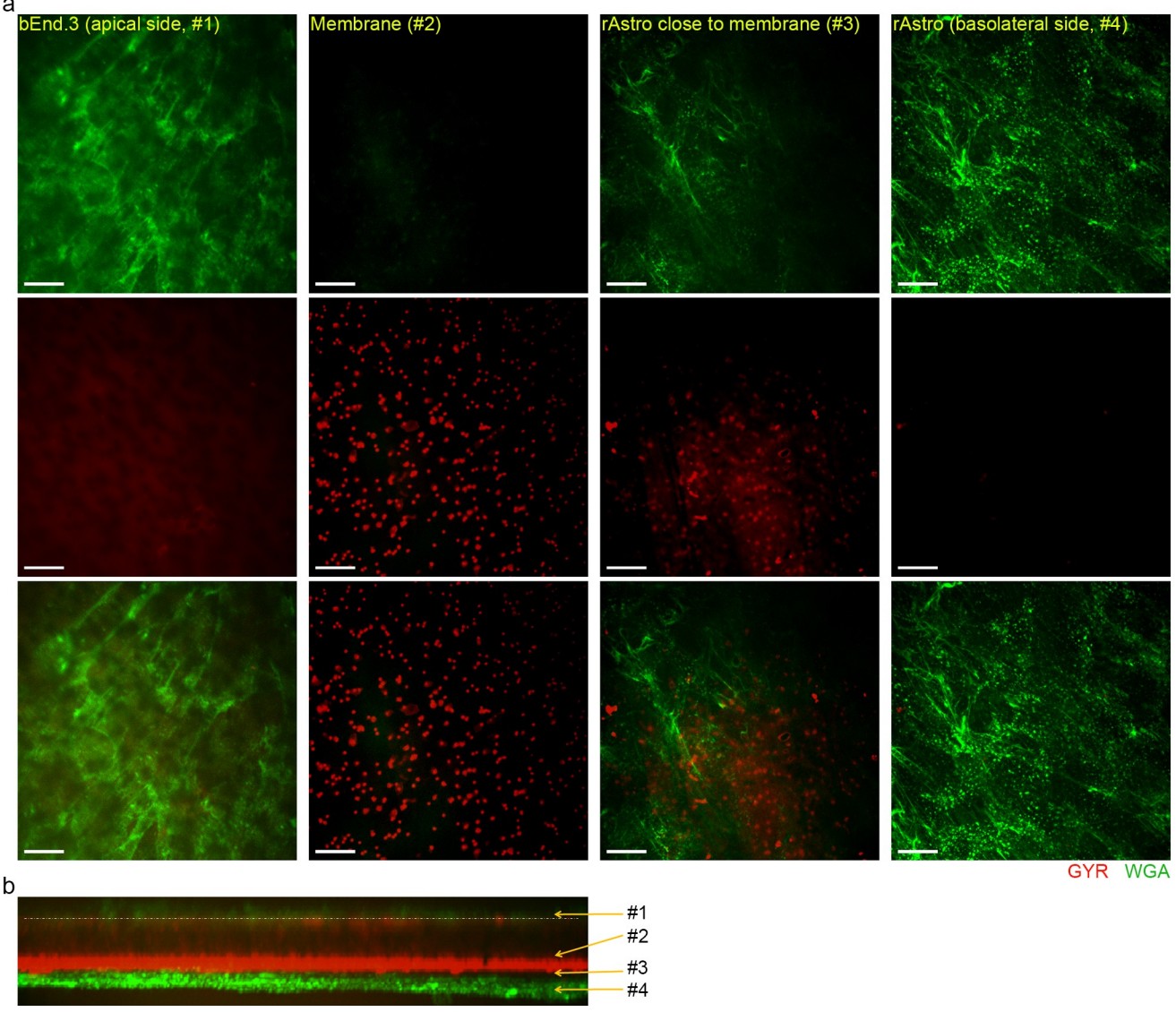

**Fig 7. Translocation of GYRp (4).** Translocation of 10 μM TAMRA-labeled GYRp across confluent bEnd.3 cells cultured as contact co-culture with rAsto (a-b) or monoculture (c). (a) Representative 2D micrographs of different regions of the co-culture show that most of the GYRp is trapped inside the membrane pores (red: GYRp, green: WGA labeled cells). Scale bars: 20 μm. (b) Cross-sectional view of the entire field-of-view. The white dashed line marks the top part of the 10 μm thick membrane. Image acquisition: 60x water immersion objective.

hCMEC/D3 cell line, where the permeability of 4 kDa Dextran is ranging between ~5–14 x 10$^{-6}$ cm/s when cultured on Transwell membrane [35–37]. Our translocation studies suggest that GYR can transcytose across the BBB. The comparison of P$_{app}$ values of cell-free filters and the imaging studies (Table 3 and S5 Fig) revealed MTfp binds to the membrane and/or is trapped inside the pores of the membrane; consequently, these methods were not suitable to quantitatively determine the translocation ability of MTfp. However, as Fig 6 indicates, MTfp is able to cross the endothelial cell layer, but it is trapped inside the membrane afterwards. The use of different type of membranes could resolve this issue as the material of the membrane, pore size, and porosity could affect the peptide penetration. Furthermore, binding of MTfp to the endocytic receptors, LRP-1 and TfR, indicates that at a least a fraction of the observed translocated MTfp is conveyed by transcellular transport. Since bEnd.3 cells form a rather leaky barrier, the

translocation studies should be repeated with a tighter *in vitro* BBB model, such as induced pluripotent stem cell-based models, or *in vivo* experiments in mice.

One aspect that has been discussed in the literature in terms of uptake and delivery has been the affinity to the targeting receptor. A high affinity to the receptor leads to efficient binding but may also lead to impaired release from the receptor and decreased transcellular transport [5, 38–40]. Thus, an intermediate binding affinity could be preferred. Interestingly, despite being developed for human targets [11, 12, 17], both MTfp and GYR bind to murine TfR, as shown by SPR. For the GYR peptide, the applied concentration suggests that other uptake mechanisms may be in place (10 μM GYR vs. a $K_D$ of 1 mM), whereas at this concentration, strong binding to the murine TfR by the MTf peptide is to be expected. However, LRP-1 and TfR are only two out of a multitude of possible transporters available, but our energy-dependent uptake experiments indicate that peptides enter the bEnd.3 cells by receptor-mediated endocytosis and not by a passive mechanism. The binding partners for GYR have not yet been identified in the literature [11, 12]. However, the submicromolar affinity to human LRP-1 indicates that this receptor might be one of them. In the literature, it has been indirectly shown that MTf is a ligand of LRP-1 [21], and GYR is a ligand of TfR and RAGE [24]. To our knowledge, the direct binding kinetics of the peptide sequences used in this study have not been investigated. From our data, it seems that a 30-minute incubation followed by a 15-minute chase stimulates a high level of uptake and transport in the endo-lysosomal pathway.

Although it has been suggested that MTf is not a ligand of TfR [21], these studies were competition experiments using Tf or the OX26 TfR antibody. Since the binding affinities of Tf or the OX26 antibody are much higher to the TfR [41] than our measured affinity of MTfp, it is plausible that in the presence of Tf and OX26, the uptake of MTf is diminished. It should be noted that the determined $K_D$ values were based on SPR binding kinetics of unlabeled peptides, and thus the affinity of the TAMRA-labeled peptides may be different from the one reported in the present SPR sensorgrams. Furthermore, the observed response in our SPR studies is much higher than what one would expect from a small peptide, indicating that both GYR and MTfp could bind to LRP-1 and TfR at multiple binding sites. Multiple binding sites combined with the fact that MTf has at least three receptor binding domains [17, 23, 42] could explain why no competition was observed with holo-transferrin in the study by Demeule *et al.* [21]. In addition, it is notable that LRP-1 in BECs has been reported to be localized at the abluminal surface [43] and that many RNA sequence databases only report very low amounts of expressed LRP-1 in these cells [44–46]. In our study, the abluminal localization was confirmed in bEnd.3 cells, and the data suggests that LRP-1 is not the main receptor facilitating the internalization of MTf to the brain *in vivo*. However, LRP-1 might have a function in the delivery of the GYR and MTfp to the abluminal surface by binding the peptides in endosomes after they have been endocytosed by TfR or other receptors. Such a handover mechanism needs to be followed up in future studies.

One factor, which challenged further studies of these peptides, was lack of fixability. Without a number of lysine residues in their amino acid sequences, aldehyde-based fixatives cannot keep the peptides in place. Nevertheless, by performing live-cell imaging, we could address whether the peptides passed the luminal glycocalyx (WGA). The lysotracker enabled us to assess the level of co-localization with lysosomes. Multiple drug delivery vehicles pass by the endo-lysosomal pathway [33, 47] and while some stay in this pathway and are degraded, others are transcytosed. A natural next step for a development platform, like the one described in this paper, is to address the level of translocation. This is outside the scope of the current work but will be of interest in future studies.

## Conclusions

In conclusion, our data quantitatively confirmed the binding of MTfp and GYR to LRP-1 and TfR, providing important knowledge for future experiments and clinical approaches using MTfp and GYR for brain optimized drug delivery. Furthermore, we studied the *in vitro* uptake and distribution of fluorescently labeled peptides in brain endothelial cells. Information on receptor binding affinity and intra-cellular transport is often lacking but is needed if one wishes to improve brain targeting of biotherapeutics.

## Supporting information

**S1 Fig. Expression of endothelial and BBB markers by bEnd.3 cells.** Confluent monolayers of bEnd.3 cells were grown on glass-bottom imaging dish and were stained for claudin-5, CD31 (PECAM-1), and ZO-1 tight and adherens junction proteins (green), and the nuclei were stained by Hoechst 33342 (blue). Scale bars: 10 μm.
(PDF)

**S2 Fig. Energy dependent uptake of TAMRA-labeled MTfp in confluent bEnd.3 cells.** The confluent bEnd.3 cells were exposed 10 μM TAMRA-labeled MTfp (red) for 1 hour at 4˚C, then the internalization of the MTfp was followed for 120 minutes (chase). Representative 2D micrographs and cross-sectional views of the regions with MTfp (yellow dashed rectangles) show that no peptide internalization happened even after 2 hours chase. MTfp was only found on the cell surface. Cells were labeled by WGA (green). Scale bars: 10 μm, image acquisition: 100x silicone immersion objective.
(PDF)

**S3 Fig. Energy dependent uptake of TAMRA-labeled GYR peptide in confluent bEnd.3 cells.** The confluent bEnd.3 cells were exposed 10 μM TAMRA-labeled GYR peptide (red, circled with magenta) for 1 hour at 4˚C, then the internalization of the peptide was followed for 120 minutes (chase). Representative 2D micrographs and cross-sectional views of the few regions with GYR peptide (yellow dashed rectangles) show that almost no peptide internalization happened even after 2 hours chase. Cells were labeled by WGA (green). Scale bars: 10 μm, image acquisition: 100x silicone immersion objective.
(PDF)

**S4 Fig. Localization of TfR and LRP-1 receptors on bEnd.3 cell surface.** Representative maximum intensity projection images and cross sectional views of the highlighted sections (yellow rectangle) show the distribution of TfR (a) and LRP-1 (b) receptors (green) on bEnd.3 cell surface. Scale bars: 10 μm, image acquisition: 100x silicone immersion objective.
(PDF)

**S5 Fig. Transcytosis control of MTfp (3) and GYRp (4).** Transcytosis of 10 μM TAMRA-labeled MTfp (a) and GYR (b) across a coated cell-free polyester Transwell membrane. (a-b) Representative 2D micrographs of the top and bottom side of the membrane and crosssectional views show that both MTfp and GYR are trapped inside the pores (red: peptides). Scale bars: 20 μm. Image acquisition: 60x water immersion objective. The white dashed line marks the top part of the 10 μm thick membrane.
(PDF)

## Acknowledgments

The authors acknowledge the AU Health Bioimaging Core Facility and AU Molecular Interactions Core Facility. Søren K. Moestrup, University of Aarhus, kindly provided purified LRP-1. Claus Pietrzik, University of Mainz, kindly provided anti-LRP-1. Furthermore, Annemette Boe Marnow, University of Aarhus, is acknowledged for her technical assistance.

## Author Contributions

**Conceptualization:** Diána Hudecz.

**Data curation:** Diána Hudecz.

**Formal analysis:** Sara Björk Sigurdardóttir, Casper Hempel.

**Funding acquisition:** Casper Hempel, Morten S. Nielsen.

**Investigation:** Sarah Christine Christensen, Casper Hempel.

**Methodology:** Diána Hudecz, Sarah Christine Christensen, Morten S. Nielsen.

**Resources:** Thomas Lars Andresen, Morten S. Nielsen.

**Supervision:** Andrew J. Urquhart, Thomas Lars Andresen, Morten S. Nielsen.

**Writing – original draft:** Diána Hudecz, Sara Björk Sigurdardóttir, Casper Hempel, Morten S. Nielsen.

**Writing – review & editing:** Casper Hempel, Morten S. Nielsen.

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
