## [Decision Letter · Decision Letter 0]

4 Nov 2020

PONE-D-20-31660

Two peptides targeting endothelial receptors are internalized into murine brain endothelial cells

PLOS ONE

Dear Dr. Nielsen,

Thank you for submitting your manuscript to PLOS ONE. After careful consideration, we feel that it has merit but does not fully meet PLOS ONE’s publication criteria as it currently stands. Therefore, we invite you to submit a revised version of the manuscript that addresses the points raised during the review process.

Three experts have evaluated the manuscript. They agreed, that the research topic is very relevant, but also that amendment is needed including some additional experiments  (regarding the specificity of the peptides eg. uptake inhibitors, 4C). The authors need to justify the selection of the two peptides and the use of the b.End3 model. 

We look forward to receiving your revised manuscript.

Kind regards,

Mária A. Deli, M.D., Ph.D.

Academic Editor

PLOS ONE

Journal Requirements:

2. Please ensure that you refer to Figures 1, 2 and 3 in your text as, if accepted, production will need this reference to link the reader to each figure.

Reviewers' comments:

Reviewer's Responses to Questions

**Comments to the Author**

1. Is the manuscript technically sound, and do the data support the conclusions?

Reviewer #1: No

Reviewer #2: Partly

Reviewer #3: Partly

2. Has the statistical analysis been performed appropriately and rigorously? 

Reviewer #1: I Don't Know

Reviewer #2: N/A

Reviewer #3: Yes

3. Have the authors made all data underlying the findings in their manuscript fully available?

Reviewer #1: Yes

Reviewer #2: No

Reviewer #3: Yes

4. Is the manuscript presented in an intelligible fashion and written in standard English?

Reviewer #1: Yes

Reviewer #2: Yes

Reviewer #3: Yes

5. Review Comments to the Author

Reviewer #1: In this manuscript, Sigurdardóttir and colleagues investigate the ability of 2 synthetic peptides to bind and to be uptaken by murine brain endothelial cells. This topic is very interesting because there is a need to identify new strategies to overcome the blood-brain barrier in order to treat neurodegenerative diseases. From my point of view, the manuscript is well written but conclusions need to be supported by more experiments and addition of controls. In particular, I would suggest to perform inhibition experiments or transcytosis experiments, and to use scramble peptides as controls.

Please find below my comments that need to be addressed:

- References need to be carefully checked because it appears in several places that the reference sources were not found (lines 240, 247, 258, 280 to 290, etc).

- As written in the conclusion, expression and localization of LRP1 and other members of the LDLR family are very confusing in the literature. To support their results, authors need to quantify these expressions and to investigate the luminal/abluminal expression of them.

- Lines 52-53, authors claim that RAP inhibits LRP1, but in fact, RAP inhibits all members of the LDLR family. Therefore, to specifically inhibit LRP1, authors should use Sh/SiRNA or mutagenesis approaches.

- Picture quality needs to be improved. To really demonstrate that the 2 peptides are specifically endocytosed by LRP/TfR, authors might perform several additional experiments including competitive experiments (using RAP or LRP antibodies), and should compare results obtained together in 4°C versus 37°C experiments. Inhibition of transyctosis pathways can be also performed using inhibitors or cyclodextrins, etc. Furthermore, scramble peptides should be designed to demonstrate that internalization process is specific, etc.

- Authors wrote in numerous places of the manuscript that they study transcytosis of these peptides but there is no experiment showing transcytosis results. Authors should re-phrase or might perform these experiments by seeding b.end3 cells on transwell inserts.

- In addition, authors wrote (line 270) that “Both peptides were detected in all horizontal planes of the BECS, also on the lower (basal) cell membrane indicating transcytosis”, but if the cells are leaky it is likely that these peptides cross the cells through the opened tight junctions and accumulate directly on the slides. In addition, if transcytosis process occurs, it could be mediated by adsorptive pathway and not necessary by receptor-mediated transcytosis. Again, adding adequate controls (scramble peptides) and performing competitive experiments might allow to generate more relevant data.

- Because these peptides seem to preferentially interact with the human LRP1 and TfR, why authors decided to use a murine in vitro BBB model instead to use human iPSC or stem cells ? This point should be discussed in the introduction/discussion parts.

Reviewer #2: In “Two peptides targeting endothelial receptors are internalized into murine brain endothelial cells”, Sigurdardóttir et al. describes the interaction of two transBBB peptides with endothelial receptors, as well as the internalization pathway followed by both peptides in brain endothelial cells. The work is well organized and clear. However, the manuscript needs extra experiments and clarifications:

Major general comments:

1. I believe the two peptides employed in this study are not clearly described, should be clear their relevance and what distinguishes the peptides from other in the literature that target the same receptors. Therefore, I would like to know more about the rationale behind the design of both peptides. I think that the authors should include information on previous results that support the inclusion of both peptides in the study, and current applications (if possible).

2. In the literature, there are numerous examples of the use of bEnd.3 cells in in vitro model of BBB to evaluate peptides translocation ability. The manuscript would benefit from those assays to prove the translocation of MTfp and GYRp? It would also be interesting to should the capability of these peptides to deliver a cargo.

3. Another concern with the experimental design is related to the use of a mouse cell line (bEnd.3) instead of a human endothelial cell line. In the manuscript the authors show that both peptides have a good binding affinity towards human receptors (LRP-1 and hTfR) and poor binding affinity towards mouse receptors (mTfR). So, based on these results and the increased interest of human models, why did you select the bEnd.3 as your cell line model?

4. Figure resolution should be improved. For instance, in figure 3 the text and sensorgrams are out of focus.

5. All references from results section are gone. “Error! Reference source not found”.

My major concern is related to the innovative aspects of the manuscript. At first, it seems that peptides are original or at least not tested for BBB interaction. However, GYRp was developed by phage isolation from the brain and MTfp was used to deliver antibodies in vivo, where biodistribution demonstrates delivery of MTfp-antibody into brain parenchyma. Both peptides were tested in mice, thus it would be pertinent to test their ability in BBB models of human brain endothelial cells. The SPR results do not add additional information, since authors state “This suggests that LRP-1 is not the main receptor facilitating the transcytosis of MTf to the brain in vivo”.

Other particular comments:

Abstract (Page 2)

Line 26 – Some issue raise above concerning the use of a murine model if both peptides have more affinity towards human receptors.

Line 30 – “frequently” is a subjective concept.

Introduction (Page 3)

The introduction presents the problem of crossing the BBB in a very clear way. It shows the need for innovative strategies and the authors try to address this problem by using two peptides.

Line 47 – For MTf a fair description is present. Nevertheless, I would like to know a little more about the physiological role of MTf.

Line 50 – The sentences are confusing.

Line 54 – Do you think that the interaction with BBB receptors might be compromised by the conjugation to small molecule drugs? Did you consider the use of conjugated peptides in the study?

Line 57 – “(…) MTf and MTf-derived peptides”, Why did you select MTfp among the other “promising as brain delivery agents?”

Line 59 – Poor description. The authors need to present GYR peptide in a more clear way.

Line 63 – “(…) of GYR and MTf-derived peptides (…)” is not a completely true sentence. The authors use one MTf-derived peptide, namely the MTfp.

Materials and Methods (Page 4)

Line 70 – It is not fluorophores. The authors only conjugated both peptide to one fluorophore (TAMRA).

Line 70 – Why did you conjugated the fluorophore in the N-terminus for one peptide and in the C-terminus for the other one?

Line 176 – “relevant secondary antibodies” …

Line 193 – Why did the authors incubated both peptides for 30 min?

Line 228 – The concentrations should be presented in the same way. There are differences between the text and the figure, for instance, (250 mM, 500 mM) and (0.25 uM, 0.5 uM).

Results (Page 13)

All references are absent.

Line 234-248 – are not results but methods.

Line 260 – Please comment the SPR sensorgrams for GYRp. Sensorgrams are not a typical binding curve with association, steady state and dissociation. The fast dissociation is typical of non-specific binding.

Line 255 – The authors mention the use of WB to confirm the TfR and LRP-1 expression in bEnd.3; however, they do not present the results. Since this section is basically based on the SPR results, and in these experiments the authors do not use cells, is important to present those results.

Line 271 – “on the lower (basal) cell membrane indicating transcytosis”- is not correct to say “transcytosis”. The term transcytosis means that the molecule to be transported is captured in vesicles on one side of the cell, drawn across the cell, and release on the other side. In the results presented there is no confirmation of release on the other side.

The authors should use other controls to demonstrate internalization, for instance, perform the experiment at 4 C.

Discussion (Page 16)

Line 300 – “(…) targeting brain diseases (…)” instead of “(…) targeting diseases (…)”.

Line 305 – A comment on the conjugation at the N- or C-terminus would be helpful to understand the difference if the authors considered it interesting.

Line 319 – With substantially less affinity. Comment on that in the discussion.

Line 321 – Why is this to be expected?

Line 331 – The authors introduce OX26 as a “control”; however, they do not explain the relevance of using it. An explanation would improve its impact.

Line 334 – If you consider that the affinity would be different, why did you not use fluorescent-labeled peptides instead of unlabeled ones?

Line 343 – “fixability”

Conclusions:

The authors claim they have developed a platform for synthesis and testing of brain-targeting peptides. Do the authors consider LRP1 and TfR brain specific receptors? Do results show strong evidence of targeting to these receptors?

“Information on receptor binding affinity and intra-cellular transport is often lacking” the authors should consider other methods to determine transport pathways. Labeling of a single compartment, such as lysosomes is not sufficient to determine “intra-cellular transport”.

Reviewer #3: In the manuscript “Two peptides targeting endothelial receptors are internalized into murine brain endothelial cells”, the authors investigate two synthetic peptides, MTfp and GYRp as BBB transcytosing peptides. Sigurdardóttir et al. demonstrate the binding ability to LRP1 and TfR receptors and the uptake and intracellular trafficking of these peptides in mouse brain endothelial cells, bEnd3. The topic of the present work is very important because transBBB peptides are promising new tools for drug delivery to the brain. The manuscript contains interesting results but in my opinion, authors need to perform extra experiments to support the conclusions.

Major comments:

1. The authors used mouse bEnd3 cell line as BBB model, however it is well known from the literature, that this model is leaky and very week as in vitro BBB model. Furthermore, the authors demonstrate that both peptides have better binding affinity towards human receptors compare to mouse ones, so authors should clarify why they selected bEnd3 cells for these experiments.

2. From the introduction part, I miss more information about the two peptides and the background results, mostly about the GYR peptides. Line 57 and 62, authors mentioned other promising transBBB peptides, why did the authors choose these ones for further studies.

3. The quality of the figures of immunocytochemistry should be improved, the text is unreadable.

4. To prove the uptake of the tested peptides in bEnd3 cells, the authors should perform extra experiments with scramble peptides as controls, investigate the temperature dependency of the cellular uptake (37°C - 4 °C) or use inhibitors.

5. The authors mentioned BBB transcytosis experiment in the abstract and later in several parts of the manuscript but there is no data demonstrating these results. Authors should show these results.

Minor comments:

1. The first part of the Results ”Peptide synthesis and fluorophore labeling” belongs to the Methods, please correct it.

2. All of the references are missing from the Results part (Error! Reference source not found.), please correct them.

In conclusion, the manuscript needs more data to underlay the authors claims.

6. PLOS authors have the option to publish the peer review history of their article (what does this mean?). If published, this will include your full peer review and any attached files.

Reviewer #1: **Yes: **Pr. Fabien Gosselet

Reviewer #2: **Yes: **Vera Neves

Reviewer #3: No

---

## [Author Response · Author response to Decision Letter 0]

22 Dec 2020

Dear Reviewers,

Thanks for the thorough review of our manuscript and the many good suggestions. We have added 6 addition figures to the manuscript (two main and 4 supplementary) with several new experiments. Below you will find a point by point response to all your comments. We hope you will find our revised version acceptable for publications in PLOS One

Sincerely

Morten Nielsen

Reviewer #1: In this manuscript, Sigurdardóttir and colleagues investigate the ability of 2 synthetic peptides to bind and to be uptaken by murine brain endothelial cells. This topic is very interesting because there is a need to identify new strategies to overcome the blood-brain barrier in order to treat neurodegenerative diseases. From my point of view, the manuscript is well written but conclusions need to be supported by more experiments and addition of controls. In particular, I would suggest to perform inhibition experiments or transcytosis experiments and to use scramble peptides as controls.

Please find below my comments that need to be addressed:

- References need to be carefully checked because it appears in several places that the reference sources were not found (lines 240, 247, 258, 280 to 290, etc).

This has been corrected. Last formatting steps with Endnote resulted in some errors we overlooked.

- As written in the conclusion, expression and localization of LRP1 and other members of the LDLR family are very confusing in the literature. To support their results, authors need to quantify these expressions and to investigate the luminal/abluminal expression of them.

We have now added new data to the manuscript. The localization and particularly the quantification at the luminal and abluminal membrane are difficult in BECs due to the thickness of the cell. As demonstrated in the new S4 Fig., with z-sections over the nucleus, we cannot observe any significant staining of LRP-1 at the luminal membrane (as observed for the TfR). These data support the published data from Zhao Z et al. (Ref 36). It has not been possible to quantify the luminal vs abluminal expression due to the limitation in the resolution of our confocal systems in the Z-axis. This important issue and consequences for the RMT of the peptides are now discussed in more details in the manuscript as well. 

- Lines 52-53, authors claim that RAP inhibits LRP1, but in fact, RAP inhibits all members of the LDLR family. Therefore, to specifically inhibit LRP1, authors should use Sh/SiRNA or mutagenesis approaches.

This is correct and a good suggestion. LRP-1 ligands (inclusive RAP) do cross-react between the LDLR receptors, and the lines in 52-54 are referring to data published in Demeule M et al. (Ref 21) – it is not our claim. We did try to inhibit the cellular uptake of both peptides with recombinant RAP but did not observe any visual differences in comparison to the uninhibited uptake (difficult to quantify). However, we did not follow up on this, as the localization of LRP-1 is mainly abluminal, and therefore we do not expect any major contribution from LRP-1 regarding luminal uptake. Speculations of a plausible function of LRP-1 in BECs is now discussed in the second last paragraph in the discussion. Due to the low amount of transcytosed peptide and the fact that transfection of BEC is difficult (and therefore difficult to knock down LRP), we did not make the siRNA knock down. 

- Picture quality needs to be improved. To really demonstrate that the 2 peptides are specifically endocytosed by LRP/TfR, authors might perform several additional experiments including competitive experiments (using RAP or LRP antibodies), and should compare results obtained together in 4°C versus 37°C experiments. Inhibition of transyctosis pathways can be also performed using inhibitors or cyclodextrins, etc. Furthermore, scramble peptides should be designed to demonstrate that internalization process is specific, etc.

We apologize for the picture quality. They are default included in the manuscript by the journal at low quality. They are made and submitted as high-quality images, and it should be possible to download and expect them by clicking at the link in Manuscript/PDF document. We have now added new data (S2 Fig. and S3 Fig.) demonstrating the energy depending uptake of the peptides and included corresponding text in the manuscript. The use of scrambled peptides is not straight forward due to the labeling approach and methods we have used. 

- Authors wrote in numerous places of the manuscript that they study transcytosis of these peptides but there is no experiment showing transcytosis results. Authors should re-phrase or might perform these experiments by seeding b.end3 cells on transwell inserts.

We have added transcytosis experiments to the manuscript, see new Fig. 6 and Fig. 7 and corresponding text. We do underline in the discussion that these data should be further demonstrated in vivo (in mice) or by iPSC, as the bEnd3 model is not completely tight. 

- In addition, authors wrote (line 270) that “Both peptides were detected in all horizontal planes of the BECS, also on the lower (basal) cell membrane indicating transcytosis”, but if the cells are leaky it is likely that these peptides cross the cells through the opened tight junctions and accumulate directly on the slides. In addition, if transcytosis process occurs, it could be mediated by adsorptive pathway and not necessary by receptor-mediated transcytosis. Again, adding adequate controls (scramble peptides) and performing competitive experiments might allow to generate more relevant data.

We have now demonstrated using 4-37 °C uptake (S2 Fig. and S3 Fig.) that the uptake most likely are energy-dependent receptor-mediated uptake, and not a result of passive diffusion. As stated above, we agree that the model is leaky and it should be verified by other models as well and particularly by in vivo experiments. We agree that scramble peptides might be a good control, but we are also aware that there are numerous peptides that binds to LRP and TfR, and we might very well get an inhibition using this approach. We think that the added 4-37 °C experiments in combination with SPR would be enough for the conclusions we have in the article. 

- Because these peptides seem to preferentially interact with the human LRP1 and TfR, why authors decided to use a murine in vitro BBB model instead to use human iPSC or stem cells? This point should be discussed in the introduction/discussion parts.

Thanks for this comment. It is a very relevant question. We do have binding to mice TfR in the manuscript (Fig. 3) as the experiments were performed at bEnd3 cells. It would have been nice to have binding to mice LRP-1, but it is not possible for us to purify this receptor from mice-placenta. Moreover, as we would like to proceed with mice in vivo experiment we found that the b.End3 cells was the best choice. We have now included these considerations in the discussion. 

Reviewer #2: In “Two peptides targeting endothelial receptors are internalized into murine brain endothelial cells”, Sigurdardóttir et al. describes the interaction of two transBBB peptides with endothelial receptors, as well as the internalization pathway followed by both peptides in brain endothelial cells. The work is well organized and clear. However, the manuscript needs extra experiments and clarifications:

Major general comments:

1. I believe the two peptides employed in this study are not clearly described, should be clear their relevance and what distinguishes the peptides from other in the literature that target the same receptors. Therefore, I would like to know more about the rationale behind the design of both peptides. I think that the authors should include information on previous results that support the inclusion of both peptides in the study, and current applications (if possible).

Indeed, a good point. We have added more text in the introduction 

2. In the literature, there are numerous examples of the use of bEnd.3 cells in in vitro model of BBB to evaluate peptides translocation ability. The manuscript would benefit from those assays to prove the translocation of MTfp and GYRp? It would also be interesting to should the capability of these peptides to deliver a cargo.

Correct. We have added some transcytosis data to the manuscript, see new Fig. 6 and Fig. 7

3. Another concern with the experimental design is related to the use of a mouse cell line (bEnd.3) instead of a human endothelial cell line. In the manuscript the authors show that both peptides have a good binding affinity towards human receptors (LRP-1 and hTfR) and poor binding affinity towards mouse receptors (mTfR). So, based on these results and the increased interest of human models, why did you select the bEnd.3 as your cell line model?

This relevant point was also raised by Reviewer 1. Good human models are just as difficult to find as mouse models. However, we would like to proceed with in vivo experiments, and therefore we decided to go with the mouse cell line in the end. 

4. Figure resolution should be improved. For instance, in figure 3 the text and sensorgrams are out of focus.

Yes, the figures in the downloaded PDF are terrible. However, they were uploaded as high-quality images and the images in the PDF is a result of the conversion made by the journal. There should be a link in the PDF file, which should give you access to the original version. 

5. All references from results section are gone. “Error! Reference source not found”.

We apologize for this. This error from EndNote should now have been corrected. 

My major concern is related to the innovative aspects of the manuscript. At first, it seems that peptides are original or at least not tested for BBB interaction. However, GYRp was developed by phage isolation from the brain and MTfp was used to deliver antibodies in vivo, where biodistribution demonstrates delivery of MTfp-antibody into brain parenchyma. Both peptides were tested in mice, thus it would be pertinent to test their ability in BBB models of human brain endothelial cells. The SPR results do not add additional information, since authors state “This suggests that LRP-1 is not the main receptor facilitating the transcytosis of MTf to the brain in vivo”.

We do of course agree with these points. These studies were based on a long-term strategy to make better constructs for drug delivery to the brain. We found the MTfp in a parent from BIOasis, but previously published work claiming that TfR is not involved in the uptake of MTf (and therefore presumably also the MTfp) but is mediated by LRP-1, was not fitting with our perception of LRP-1 expression and localization in BEC. Therefore, if we should use MTfp for our future strategies to make complex dual targeting constructs, we would like to test which receptors might capable of luminal endocytosis in BEC. We think that our main conclusion, that TfR does bind MTfp and that the peptide undergoes the receptor-mediated uptake, are important for the society. Although debated, it has also demonstrated in the literature that low affinity to TfR might be important for efficient transcytosis due to the possibility that TfR might not be transported all the way to the abluminal membrane. Therefore, LRP-1 binding might still be important, since it could be involved in the exocytosis at the abluminal membrane. A final comment to this relevant discussion is that only entire MTf protein has been tested for binding to receptors before. To understand how the peptide works, binding studies (with e.g. SRP) are relevant. The GYR peptide was initially included as a positive control to our work. 

In conclusion, we agree that our results are only small bricks in a large puzzle, but we find them relevant and suitable for publication in Plos One. We hope you agree. 

Other particular comments:

Abstract (Page 2)

Line 26 – Some issue raise above concerning the use of a murine model if both peptides have more affinity towards human receptors.

Discussed above

Line 30 – “frequently” is a subjective concept.

Corrected

Introduction (Page 3)

The introduction presents the problem of crossing the BBB in a very clear way. It shows the need for innovative strategies and the authors try to address this problem by using two peptides.

Line 47 – For MTf a fair description is present. Nevertheless, I would like to know a little more about the physiological role of MTf.

Not much is known about MTf in the BBB, but we have added some text. 

Line 50 – The sentences are confusing.

Corrected.

Line 54 – Do you think that the interaction with BBB receptors might be compromised by the conjugation to small molecule drugs? Did you consider the use of conjugated peptides in the study?

It is a highly relevant question. But as we do not have antibodies against the MTfp, this is the best alternative. For transcytosis (but not for imaging), we could have used radiolabeled peptides, but new regulations in Denmark and many other countries, unfortunately, complicates this approach. 

Line 57 – “(…) MTf and MTf-derived peptides”, Why did you select MTfp among the other “promising as brain delivery agents?”

We (as many others) aim to make new dual-targeting constructs, and are therefore trying to collect several peptides with the capacity to cross the BBB. We find the MTf and MTfp are highly “underinvestigated” and might have some potential. We recently also published a paper about self-penetrating peptides (PMID: 32674358). 

Line 59 – Poor description. The authors need to present GYR peptide in a more clear way.

More explanation to GYR in the second last paragraph of the introduction has been added. 

Line 63 – “(…) of GYR and MTf-derived peptides (…)” is not a completely true sentence. The authors use one MTf-derived peptide, namely the MTfp.

Corrected to singular.

Materials and Methods (Page 4)

Line 70 – It is not fluorophores. The authors only conjugated both peptide to one fluorophore (TAMRA).

Corrected.

Line 70 – Why did you conjugated the fluorophore in the N-terminus for one peptide and in the C-terminus for the other one?

We wanted to use the same type of chemistry to label the peptides. Thus, the NHS ester is reacted with a primary amine in both peptides forming an amide binding. Since the amino acid sequence is different in the two peptides, labelling end was also not similar.

Line 176 – “relevant secondary antibodies” …

Secondary antibodies added to the table

Line 193 – Why did the authors incubated both peptides for 30 min?

We tried to visualize the uptake of the peptides in a continuous manner, i.e. visualization during continuous exposure starting from 0 min up to 60 min; however, the background from the peptide solution made the image acquisition difficult. Five to 15 min peptide incubation did not result in significant peptide internalization, i.e. they were visually not present; however, after a 30 min exposure time a significant receptor-mediated endocytosis occurred and it was possible to visually detect them. In other studies in the literature, peptide exposure varies between a few minutes and up to 60 min or even up to 24 hours.

Line 228 – The concentrations should be presented in the same way. There are differences between the text and the figure, for instance, (250 mM, 500 mM) and (0.25 uM, 0.5 uM).

Corrected.

Results (Page 13)

All references are absent.

Line 234-248 – are not results but methods.

Since the peptides were synthesized and labeled in-house (which required a long optimization process and have not been published previously), we wanted to include the synthesis and labeling as part of the result section. We understand that in some papers it is not common, but some readers with medicinal chemistry background might may find it relevant

Line 260 – Please comment the SPR sensorgrams for GYRp. Sensorgrams are not a typical binding curve with association, steady state and dissociation. The fast dissociation is typical of non-specific binding.

Text added

Line 255 – The authors mention the use of WB to confirm the TfR and LRP-1 expression in bEnd.3; however, they do not present the results. Since this section is basically based on the SPR results, and in these experiments the authors do not use cells, is important to present those results.

The WB displays the presence of the receptors in the cells. We do not find it important for the discussion of the binding. The expression is also described in the literature. Below are our blots (Figure is only in uploaded responce and can be found in end of this PDF)

Line 271 – “on the lower (basal) cell membrane indicating transcytosis”- is not correct to say “transcytosis”. The term transcytosis means that the molecule to be transported is captured in vesicles on one side of the cell, drawn across the cell, and release on the other side. In the results presented there is no confirmation of release on the other side.

The authors should use other controls to demonstrate internalization, for instance, perform the experiment at 4 C.

Several new experiments have been added

Discussion (Page 16)

Line 300 – “(…) targeting brain diseases (…)” instead of “(…) targeting diseases (…)”.

Done

Line 305 – A comment on the conjugation at the N- or C-terminus would be helpful to understand the difference if the authors considered it interesting.

This was in part chosen since we wanted to use the same fluorophore and thus not have this difference as a confounding factor. Thus, we have not included a discussion of N vs C terminal conjugation.

Line 319 – With substantially less affinity. Comment on that in the discussion.

Very low affinity is not desirable as the target (peptide, antibody etc.) needs to bind the cellular receptor for efficient uptake. In general, the binding affinity found in the literature varies between 0.4 nM to 1000 nM. 

Line 321 – Why is this to be expected?

The binding affinity of GYR to mouse TfR was only 1 mM, which is considered to be too low for successful RMT delivery. Such a low binding affinity (KD of 810 nM) could lead to successful brain delivery; however, it was reported for bispecific antibodies by Genentech (ref 5 - Yu et al. 2014, doi:10.1126/scitranslmed.3009835); they used an antibody construct with TfR and BACE1 binding. In the same year, Roche claimed that low binding affinity was needed for successful delivery (ref 7 – Niewoehner et al. 2014, doi:10.1016/j.neuron.2013.10.061)- they used monovalent antibody fragment (sFab).

Hence, we expect that GYR has a binding affinity to other receptors as well. On the other hand, we expect a stronger binding between MTfp and mouse TfR as the affinity was much lower (KD of 1.3 µM). We slightly changed the wording in the text and use “stronger” instead of “strong”.

Line 331 – The authors introduce OX26 as a “control”; however, they do not explain the relevance of using it. An explanation would improve its impact.

Text added

Line 334 – If you consider that the affinity would be different, why did you not use fluorescent-labeled peptides instead of unlabeled ones?

Yes, it could have been done. We decided to use only one constructs to get comparable results.

Line 343 – “fixability”

It is in the English dictionary. But we agree, it is not used often so we changed the text. 

Conclusions:

The authors claim they have developed a platform for synthesis and testing of brain-targeting peptides. Do the authors consider LRP1 and TfR brain specific receptors? Do results show strong evidence of targeting to these receptors?

“Information on receptor binding affinity and intra-cellular transport is often lacking” the authors should consider other methods to determine transport pathways. Labeling of a single compartment, such as lysosomes is not sufficient to determine “intra-cellular transport”.

Yes, we agree. The main problem is that we cannot fixate the peptides, and therefore the use of vesicular specific antibodies and ordinary immunofluorescence staining is not possible. We are only aware of live tracing markers for lysosomes and mitochondria, and the latter is not relevant. We are also aware that we could have tried the CellLight™ BacMam 2.0 family for staining further intracellular compartments; however, this “staining” is based on a 24 hours transfection and since we had difficulties with efficient transfection of BECs, we decided not to proceed with this technology. 

Reviewer #3: In the manuscript “Two peptides targeting endothelial receptors are internalized into murine brain endothelial cells”, the authors investigate two synthetic peptides, MTfp and GYRp as BBB transcytosing peptides. Sigurdardóttir et al. demonstrate the binding ability to LRP1 and TfR receptors and the uptake and intracellular trafficking of these peptides in mouse brain endothelial cells, bEnd3. The topic of the present work is very important because transBBB peptides are promising new tools for drug delivery to the brain. The manuscript contains interesting results but in my opinion, authors need to perform extra experiments to support the conclusions.

Major comments:

1. The authors used mouse bEnd3 cell line as BBB model, however it is well known from the literature, that this model is leaky and very week as in vitro BBB model. Furthermore, the authors demonstrate that both peptides have better binding affinity towards human receptors compare to mouse ones, so authors should clarify why they selected bEnd3 cells for these experiments.

This is relevant and raised by the other referees as well. We aim to follow up this in vitro study with in vivo experiments in mice, and therefore we decided to perform our in vitro experiments with the same species. Second, the available human models (particularly the hCMEC/D3) are just as leaky as b.End3 cells, and the alternative iPSC models have several other concerns of being more epithelial-like than endothelial. 

2. From the introduction part, I miss more information about the two peptides and the background results, mostly about the GYR peptides. Line 57 and 62, authors mentioned other promising transBBB peptides, why did the authors choose these ones for further studies.

More information on the peptide has been added. There are many interesting and promising peptides that can and should be investigated. We think the MTf peptides could be a peptide that can complement other BBB crossing peptides, particularly in a dual-targeting strategy. Therefore, we think it is relevant to study the MTfp as well.

3. The quality of the figures of immunocytochemistry should be improved, the text is unreadable.

The images are all high quality, but the formatting done by the journal resulted in poor quality images in the downloaded PDF. To get the original images, they should be downloaded separately. We apologize, but we can not do this differently. 

4. To prove the uptake of the tested peptides in bEnd3 cells, the authors should perform extra experiments with scramble peptides as controls, investigate the temperature dependency of the cellular uptake (37°C - 4 °C) or use inhibitors.

Done

5. The authors mentioned BBB transcytosis experiment in the abstract and later in several parts of the manuscript but there is no data demonstrating these results. Authors should show these results.

Done

Minor comments:

1. The first part of the Results ”Peptide synthesis and fluorophore labeling” belongs to the Methods, please correct it.

Since the peptides were synthesized and labeled in-house (which required a long optimization process and have not been published previously), we wanted to include the synthesis and labeling as part of the result section. We understand that in some papers it is not common, but some reader with medicinal chemistry background might find it relevant

2. All of the references are missing from the Results part (Error! Reference source not found.), please correct them.

Sorry, formatting issue. Should be corrected. 

In conclusion, the manuscript needs more data to underlay the authors claims.

Several new data have been included to improve the manuscript conclusions.

---

## [Decision Letter · Decision Letter 1]

25 Jan 2021

PONE-D-20-31660R1

Two peptides targeting endothelial receptors are internalized into murine brain endothelial cells

PLOS ONE

Dear Dr. Nielsen,

Thank you for submitting your manuscript to PLOS ONE. After careful consideration, we feel that it has merit but does not fully meet PLOS ONE’s publication criteria as it currently stands. Therefore, we invite you to submit a revised version of the manuscript that addresses all the points raised during the review process.

The manuscript has been greatly improved and many of the original concerns were addressed. A couple of requests remained to be answered, especially the calculation of the transcytosed peptide as a permeability coefficient, either as Papp or Pe. Another important point is the characterization of the model in terms of paracellular permeability using a marker molecule with the same MW as the peptide, eg. fluorsecently labele dextran. All the other points are minor that can be easily amended.

We look forward to receiving your revised manuscript.

Kind regards,

Mária A. Deli, M.D., Ph.D.

Academic Editor

PLOS ONE

Reviewers' comments:

Reviewer's Responses to Questions

**Comments to the Author**

1. If the authors have adequately addressed your comments raised in a previous round of review and you feel that this manuscript is now acceptable for publication, you may indicate that here to bypass the “Comments to the Author” section, enter your conflict of interest statement in the “Confidential to Editor” section, and submit your "Accept" recommendation.

Reviewer #1: (No Response)

Reviewer #2: All comments have been addressed

Reviewer #3: All comments have been addressed

2. Is the manuscript technically sound, and do the data support the conclusions?

Reviewer #1: Partly

Reviewer #2: Partly

Reviewer #3: Yes

3. Has the statistical analysis been performed appropriately and rigorously? 

Reviewer #1: I Don't Know

Reviewer #2: N/A

Reviewer #3: Yes

4. Have the authors made all data underlying the findings in their manuscript fully available?

Reviewer #1: Yes

Reviewer #2: Yes

Reviewer #3: Yes

5. Is the manuscript presented in an intelligible fashion and written in standard English?

Reviewer #1: Yes

Reviewer #2: Yes

Reviewer #3: Yes

6. Review Comments to the Author

Reviewer #1: Authors partially replied to my concerns, that has, from my opinion, improved the quality of the results and the manuscript. However, some parts can still be improved, in particular the transcytosis experiments :

- Authors claim for the MTFp that 103 % of the peptide are still in the apical compartment at the end of the experiment, whereas 3 % have transcytosed and that the rest is trapped in the coated insert. This kind of data is no really relevant and raise doubt about the methods and the quantification methods. I would suggest to the authors to use the clearance principle to calculate the MTFp permeability (Pe) across the cells. For this, I would suggest to refer to the method to generate a concentration-independent parameter as described by Siflinger-Birnboim et al. (A. Siflinger-Birnboim, P.J. Del Vecchio, J.A. Cooper, F.A. Blumenstock, J.M. Shepard, A.B. Malik, Molecular sieving characteristics of the cultured endothelial monolayer, J Cell Physiol, 132 (1987). 111-117. 10.1002/jcp.1041320115). At the end, with this method, the authors will be also able to calculate the % of recovery. Difference between the starting quantity of peptide and the remaining quantity will correspond to the part trapped in inserts and/or degraded by BEC. Because these peptides are also observed in lysosomes, this latter point might be then discussed in the manuscript. Authors are also encouraged to read this review on transport assessment when using in vitro BBB models : ”Santa-Maria AR, Heymans M, Walter FR, Culot M, Gosselet F, Deli MA, Neuhaus W. Transport Studies Using Blood-Brain Barrier In Vitro Models: A Critical Review and Guidelines. Handb Exp Pharmacol. 2020 Oct 11.”

- Then, to justify that this model, in this condition, is suitable for permeability studies, authors should include a paracellular marker such as a 3kDa-Dextran control, with an almost similar molecular weight than the transcytosed peptides of interest. Then permeabilities of the peptides and dextran need to be compared together.

- Authors wrote in the conclusion : “In conclusion, we have designed a platform allowing for the synthesis and testing of BBB-targeting peptides.”. I am not convinced that this is the main message of the manuscript and that this is really true. In addition, this kind of BBB-targeting system already exist because several labs around the world make this transcytosis or paracellular experiments, routinely, even with human BBB models that are often more tight and easy to handle than b.end3 cells.

Reviewer #2: The authors have addressed the main concerns pointed out in the review report. Thus, the quality of the manuscript increased substantiality. However, I still have some concerns related to the translocation assays and the respective statements.

The authors followed the reviewers’ suggestion and introduced a translocation assay. This is a big and important part of the work, since these peptides were designed to penetrate the brain. The model used is a co-culture, which is usually referred as an improvement to single cell models.

Major comments I would like to be addressed:

1. Was the integrity of barrier measured, using a fluorescent probe?

2. Why did you use microscopy to detect translocation? You could have detect fluorescence in the basolateral compartment by measuring the fluorescence intensity of labeled-peptides.

3. You should check stability of the peptides. Are peptides degraded in acidic conditions?

4. Can 2.9 and 3.7% translocation be considered good translocation? Have you determine a threshold for translocation in your BBB models? In my view, the authors cannot state that MTfp and GYR “transcytose across the BBB”. The authors should rephrase the following sentences:

Line 31 – “Moreover, our in vitro Transwell transcytosis experiments confirmed that both MTfp and GYR peptides were able to transcytose across a murine barrier. Thus, despite binding to endocytic receptors with different affinities, both peptides are able to transcytose across the murine BECs.”

Line 398 – “Our transcytosis studies suggest that both peptides can transcytose across the BBB.

5. What do the authors mean with “The rest were most likely trapped inside the membranes.”? Since the results are “2.9±0.8% MTfp and 3.7±0.2% GYR were able to transcytose into the basolateral compartment, whereas 103.1±2.6% and 94.6±0.5% remained inside the apical compartment” is there something left?

Minor comments:

Line 20 – conjugating instead of attaching

Line 56 – I believe reference 21 is not correct, as Demeule et al 2002 does not use doxorubicin. Please check all references.

Line 182 – lacks a for five minutes

Line 224 – Lacks a “(“

Reviewer #3: The authors have adequately addressed my comments and added several new data to the manuscript. They tested both the temperature dependency of the cellular uptake and the transcytosis of the peptides upon my request.

I have only one minor finding: the transcytosis of MTfp and GYR peptides would be better to show with the calculation of the apparent permeability coefficient and not with the form of % of transcytosed peptide.

7. PLOS authors have the option to publish the peer review history of their article (what does this mean?). If published, this will include your full peer review and any attached files.

Reviewer #1: **Yes: **Fabien Gosselet

Reviewer #2: **Yes: **Vera Neves

Reviewer #3: No

---

## [Author Response · Author response to Decision Letter 1]

5 Mar 2021

The manuscript has been greatly improved and many of the original concerns were addressed. A couple of requests remained to be answered, especially the calculation of the transcytosed peptide as a permeability coefficient, either as Papp or Pe. Another important point is the characterization of the model in terms of paracellular permeability using a marker molecule with the same MW as the peptide, eg. fluorsecently labele dextran. All the other points are minor that can be easily amended.

Dear Editor and reviewers,

Thanks for the many nice comments and suggestions. We have now studied the permeability in more detail and evaluated the translocation using Papp and Pe. As discussed by you and our comments in the last rebuttal, the bEnd.3 cells are not the best model to study transcytosis. Our major message in this publication is the binding and endocytosis of the peptides to endocytic receptors, which is the first step in a successful transcytosis from blood to brain. The transcytosis/translocation studies we have provided is indicative and should be further validated in vivo. This should hopefully be clear from the discussion and conclusion. Below we have addressed all concerns and comments point-by-point. We hope that the manuscript is acceptable now for publication in PlosOne.

Sincerely 

Morten Nielsen

Reviewer #1: Authors partially replied to my concerns, that has, from my opinion, improved the quality of the results and the manuscript. However, some parts can still be improved, in particular the transcytosis experiments :

- Authors claim for the MTFp that 103 % of the peptide are still in the apical compartment at the end of the experiment, whereas 3 % have transcytosed and that the rest is trapped in the coated insert. This kind of data is no really relevant and raise doubt about the methods and the quantification methods. I would suggest to the authors to use the clearance principle to calculate the MTFp permeability (Pe) across the cells. For this, I would suggest to refer to the method to generate a concentration-independent parameter as described by Siflinger-Birnboim et al. (A. Siflinger-Birnboim, P.J. Del Vecchio, J.A. Cooper, F.A. Blumenstock, J.M. Shepard, A.B. Malik, Molecular sieving characteristics of the cultured endothelial monolayer, J Cell Physiol, 132 (1987). 111-117. 10.1002/jcp.1041320115). At the end, with this method, the authors will be also able to calculate the % of recovery. Difference between the starting quantity of peptide and the remaining quantity will correspond to the part trapped in inserts and/or degraded by BEC. Because these peptides are also observed in lysosomes, this latter point might be then discussed in the manuscript. Authors are also encouraged to read this review on transport assessment when using in vitro BBB models : ”Santa-Maria AR, Heymans M, Walter FR, Culot M, Gosselet F, Deli MA, Neuhaus W. Transport Studies Using Blood-Brain Barrier In Vitro Models: A Critical Review and Guidelines. Handb Exp Pharmacol. 2020 Oct 11.”

As suggested, the translocation was recalculated using the clearance principle and is presented as apparent and effective permeability (Papp and Pe). To calculate the apparent permeability, we used the following equation based on Czupalla et al. (doi: 10.1007/978-1-4939-0320-7_34):

P_app [cm/s]=B/T∙V_b/(A∙t∙60) 

Where Papp is the apparent permeability, B is the relative fluorescence unit (RFU) at time t (120 min), T is the top chamber RFU at time 0 (assumed to be constant; hence, we used the top chamber RFU at 120 min), Vb is the volume of the bottom channel [ml], A is the cross-section area of the membrane [cm2], and t is the time [min]. The reason for using this equation instead of the suggested V=(〖[A]〗_b∙V_b)/〖[A]〗_t and P_app=dV/(dt∙A) (where [A]b is the concentration of the tracer in the basolateral compartment and [A]t is the initial concentration of the tracer in the basolateral/top compartment) from Siflinger-Birnboim et al. (doi: 10.1002/jcp.1041320115) is that the former equation is entirely based on the measured fluorescence intensity and not on the calculated concentrations based on a standard curve, i.e. in our opinion, a slightly bit more accurate. Furthermore, we calculated the permeability coefficients using both equations and the results were very similar. In the case of GYR-TAMRA, the Papp were 6.64 x10-6 cm/s vs. 6.71 x10-6 cm/s vs. 6.31 x10-6 cm/s using RFU values, the calculated concentrations based on a standard curve or the calculated final basal concentration and the 10 µM initial apical concentration values, respectively. The new data has been included in the manuscript as Table 3.

 - Then, to justify that this model, in this condition, is suitable for permeability studies, authors should include a paracellular marker such as a 3kDa-Dextran control, with an almost similar molecular weight than the transcytosed peptides of interest. Then permeabilities of the peptides and dextran need to be compared together.

We used 4 kDa Dextran (FD4 from Sigma, molecular weight (MW) is ranging between 3 and 5 kDa). Both Papp and Pe was calculated and included in the manuscript and the values were compared with each other. The Papp through the cell-free filter revealed that most of the MTfp are trapped inside the membrane (as it was confirmed by microscopy) as the Papp, filter, MTfp was smaller than Papp, filter, Dextran. Since the MW of the labelled MTfp is smaller than the dextran (~2.6 kDa vs. ~4 kDa), the expected Papp, filter, MTfp is higher than Papp, filter, Dextran as it was seen in the case of GYR. Hence, Transwell membrane (or at least polyester membrane with 0.4 µm pore size) and fluorescence intensity-based analysis of the basolateral compartment are not suitable to measure the translocation of MTfp across the BBB. This part is included and explained in the manuscript as well. 

All permeability values are presented in Table 3.

- Authors wrote in the conclusion : “In conclusion, we have designed a platform allowing for the synthesis and testing of BBB-targeting peptides.”. I am not convinced that this is the main message of the manuscript and that this is really true. In addition, this kind of BBB-targeting system already exist because several labs around the world make this transcytosis or paracellular experiments, routinely, even with human BBB models that are often more tight and easy to handle than b.end3 cells.

Thank you, we agree with you and it has been modified in the manuscript. 

Reviewer #2: The authors have addressed the main concerns pointed out in the review report. Thus, the quality of the manuscript increased substantiality. However, I still have some concerns related to the translocation assays and the respective statements.

The authors followed the reviewers’ suggestion and introduced a translocation assay. This is a big and important part of the work, since these peptides were designed to penetrate the brain. The model used is a co-culture, which is usually referred as an improvement to single cell models.

Major comments I would like to be addressed:

1. Was the integrity of barrier measured, using a fluorescent probe?

The barrier integrity was determined by measuring the paracellular permeability of fluorescein isothiocyanate (FITC) labeled 4 kDa dextran and sodium fluorescein (376 Da) and these data are now included in the manuscript.

2. Why did you use microscopy to detect translocation? You could have detect fluorescence in the basolateral compartment by measuring the fluorescence intensity of labeled-peptides.

We used microscopy and traditional fluorescence intensity-based techniques to detect translocation. We used microscopy as we wanted to see if the peptides were able to be endocytosed and translocate through the membrane and taken up by the astrocytes grown in juxtaposition on the opposite side of the Transwell membrane. Binding to endocytic receptors is a critical point for further success in in vivo translocation and successful drug delivery. Although we did not observe any translocated peptides inside the astrocytes using microscopy (most of them trapped inside the pores of the polyester membrane), a degree of translocation was observed by fluorescence intensity measurement of the basolateral compartment. 

3. You should check stability of the peptides. Are peptides degraded in acidic conditions?

This is a valid request. We initially labeled a large fraction of peptides with TAMRA, which subsequently was purified and validated using HPLC in smaller portions when needed. The purification over time did not indicate any degradation but we can unfortunately not provide any quantitative data.

4. Can 2.9 and 3.7% translocation be considered good translocation? Have you determine a threshold for translocation in your BBB models? In my view, the authors cannot state that MTfp and GYR “transcytose across the BBB”. The authors should rephrase the following sentences:

Line 31 – “Moreover, our in vitro Transwell transcytosis experiments confirmed that both MTfp and GYR peptides were able to transcytose across a murine barrier. Thus, despite binding to endocytic receptors with different affinities, both peptides are able to transcytose across the murine BECs.”

Line 398 – “Our transcytosis studies suggest that both peptides can transcytose across the BBB.

The translocation was recalculated and is now presented as apparent and effective permeability. These permeability values were compared with the permeability of 4 kDa dextran, and they are discussed in the manuscript. 

We agree that we have made conclusions about transcytosis mechanisms that are not proved by our data. Therefore, the term “trancytosis” is now corrected to “translocation” in all relevant places in the manuscript text.

5. What do the authors mean with “The rest were most likely trapped inside the membranes.”? Since the results are “2.9±0.8% MTfp and 3.7±0.2% GYR were able to transcytose into the basolateral compartment, whereas 103.1±2.6% and 94.6±0.5% remained inside the apical compartment” is there something left?

As our translocation data is now presented as permeability (see Table 3), and the referred texts are now removed from the text. 

However, we meant that in the case of the GYR, we collected 3.7 + 94.6 = 98.3% < 100%, and the remaining 1.7% is most likely trapped inside the filter membrane or in the endothelial cell layer. Again, we would like to stress that the receptor identification and binding is the most important message of the manuscript. Therefore the conclusion on translocation is less strong at the moment. 

Minor comments:

Line 20 – conjugating instead of attaching

Line 56 – I believe reference 21 is not correct, as Demeule et al 2002 does not use doxorubicin. Please check all references.

Line 182 – lacks a for five minutes

Line 224 – Lacks a “(“

Many thanks for these observations, they are now corrected in the text.

Reviewer #3: The authors have adequately addressed my comments and added several new data to the manuscript. They tested both the temperature dependency of the cellular uptake and the transcytosis of the peptides upon my request.

I have only one minor finding: the transcytosis of MTfp and GYR peptides would be better to show with the calculation of the apparent permeability coefficient and not with the form of % of transcytosed peptide.

Thank you very much. The translocation of the peptides is now presented as apparent permeability coefficient. Furthermore, the calculated effective permeability coefficient values were also calculated (see Table 3).

Furthermore, following the statistical analysis guidelines of PLOS One, our data is presented as mean (SD) instead of mean ± SD. Furthermore, the normality of data distribution was assessed with GraphPad Prism 9.0; the QQ plots can be found below. These plots are not included in the manuscript.

---

## [Decision Letter · Decision Letter 2]

23 Mar 2021

Two peptides targeting endothelial receptors are internalized into murine brain endothelial cells

PONE-D-20-31660R2

Dear Dr. Nielsen,

We’re pleased to inform you that your manuscript has been judged scientifically suitable for publication and will be formally accepted for publication once it meets all outstanding technical requirements.

Kind regards,

Mária A. Deli, M.D., Ph.D.

Academic Editor

PLOS ONE

Additional Editor Comments (optional):

Reviewers' comments:

Reviewer's Responses to Questions

**Comments to the Author**

1. If the authors have adequately addressed your comments raised in a previous round of review and you feel that this manuscript is now acceptable for publication, you may indicate that here to bypass the “Comments to the Author” section, enter your conflict of interest statement in the “Confidential to Editor” section, and submit your "Accept" recommendation.

Reviewer #1: All comments have been addressed

Reviewer #2: All comments have been addressed

Reviewer #3: All comments have been addressed

2. Is the manuscript technically sound, and do the data support the conclusions?

Reviewer #1: Yes

Reviewer #2: Yes

Reviewer #3: Yes

3. Has the statistical analysis been performed appropriately and rigorously? 

Reviewer #1: I Don't Know

Reviewer #2: Yes

Reviewer #3: Yes

4. Have the authors made all data underlying the findings in their manuscript fully available?

Reviewer #1: Yes

Reviewer #2: Yes

Reviewer #3: Yes

5. Is the manuscript presented in an intelligible fashion and written in standard English?

Reviewer #1: Yes

Reviewer #2: Yes

Reviewer #3: Yes

6. Review Comments to the Author

Reviewer #1: (No Response)

Reviewer #2: In “Two peptides targeting endothelial receptors are internalized into murine brain endothelial cells”, Hudecz et al. describe the interaction of two BBB translocation peptides with endothelial receptors, as well as the internalization pathway and translocation efficiency of both peptides in brain endothelial cells. The authors have addressed the main concerns pointed out in previous review report and I endorse its publication.

Reviewer #3: The authors have adequately addressed my comments raised in the previous round of review and I feel that this manuscript is now acceptable for publication.

7. PLOS authors have the option to publish the peer review history of their article (what does this mean?). If published, this will include your full peer review and any attached files.

Reviewer #1: **Yes: **Pr. Fabien Gosselet

Reviewer #2: No

Reviewer #3: No

---

## [Editor Report · Acceptance letter]

25 Mar 2021

PONE-D-20-31660R2 

Two peptides targeting endothelial receptors are internalized into murine brain endothelial cells 

Dear Dr. Nielsen:

I'm pleased to inform you that your manuscript has been deemed suitable for publication in PLOS ONE. Congratulations! Your manuscript is now with our production department. 

Kind regards, 

on behalf of

Dr. Mária A. Deli 

Academic Editor

PLOS ONE